# Regulation of acetyl-CoA biosynthesis via an intertwined acetyl-CoA synthetase/acetyltransferase complex

Liujuan Zheng [1,2,5] ✉, Yifei Du [3,5], Wieland Steinchen[2], Mathias Girbig [1], Frank Abendroth[2], Ekaterina Jalomo-Khayrova [2], Patricia Bedrunka[2], Isabelle Bekeredjian-Ding[4], Christopher-Nils Mais[2], Georg K. A. Hochberg [1,2], Johannes Freitag [2] & Gert Bange [1,2] ✉

Acetyl-CoA synthetase (Acs) generates acetyl-coenzyme A (Ac-CoA) but its excessive activity can deplete ATP and lead to a growth arrest. To prevent this, Acs is regulated through Ac-CoA-dependent feedback inhibition executed by Ac-CoA-dependent acetyltransferases such as AcuA in *Bacillus subtilis*. AcuA acetylates the catalytic lysine of AcsA turning the synthetase inactive. Here, we report that AcuA and AcsA form a tightly intertwined complex – the C-terminal domain binds to acetyltransferase domain of AcuA, while the C-terminus of AcuA occupies the CoA-binding site in the N-terminal domain of AcsA. Formation of the complex reduces AcsA activity in addition to the well-established acetylation of the catalytic lysine 549 in AcsA which we show can disrupt the complex. Thus, different modes of regulation accomplished through AcuA adjust AcsA activity to the concentrations of the different substrates of the reaction. In summary, our study provides detailed mechanistic insights into the regulatory framework underlying acetyl-CoA biosynthesis from acetate.

Cells need to continuously adapt their metabolism to changing qualities and amounts of nutrients. The "acetate switch" is a well-known example that occurs as cells exhaust their environment's acetate-producing carbon sources (e.g., glucose), prompting them to scavenge environmental acetate as alternative carbon source (reviewed in ref. 1). The term "acetate switch" specifies the metabolic adaptability of cells to either excrete or absorb acetate depending on nutrient availability and was defined as the point where both processes are in equilibrium[1,2]. Central to the "acetate switch" is the conversion of acetate into its activated derivative acetyl-Coenzyme A (Ac-CoA)[2–4].

This reaction is catalyzed by Ac-CoA synthetase (Acs, acetate:CoA ligase, EC 6.2.1.1.). Initially, acetate and ATP are transformed in an acetyl-adenylate intermediate (Ac-AMP). Subsequently, this intermediate serves as a donor for the acetyl group to the sulfhydryl group of CoA, yielding Ac-CoA as the final product[3]. Acs enzymes consists of two consecutively operating modules, a C-terminal domain (CTD) catalyzing the first and a N-terminal domain (NTD) for the second reaction step[3,5,6]. In addition to Acs, an alternative pathway for acetate absorption with lower affinity has been identified. This pathway involves acetate kinase converting acetate to acetylphosphate (AcP), followed by catalysis of Ac-CoA acetyltransferase to generate Ac-CoA[7]. This process primarily enables both acetate absorption and excretion[8]. While the "acetate switch" is mostly studied in bacteria[1,9], recent research has uncovered a similar phenomenon in mammalian metabolism, e.g. in tumor cell development and stress erythropoiesis. These findings reveal Acs enzymes as potential therapeutic targets for cancer and other diseases[10–15].

[1]Max-Planck Institute for Terrestrial Microbiology, Marburg, Germany. [2]University of Marburg, Center for Synthetic Microbiology (SYNMIKRO) & Departments of Chemistry and Biology, Marburg, Germany. [3]MRC Laboratory of Molecular Biology, Cambridge, UK. [4]University of Marburg, Faculty of Medicine, Marburg, Germany. [5]These authors contributed equally: Liujuan Zheng, Yifei Du. ✉e-mail: Liujuan.Zheng@mpi-marburg.mpg.de; gert.bange@synmikro.uni-marburg.de

Overactive Acs leads to the accumulation of AMP (or ADP) and depletion of ATP, which can be fatal for the cells[16]. Consequently, Acs activity needs to be tightly regulated, which can be achieved through reversible acetylation of its catalytic lysine[17–21]. A well-studied example is Ac-CoA synthetase (AcsA) from *Bacillus subtilis*, in which lysine 549 of AcsA can be acetylated by the GCN5-related *N*-acetyltransferase (GNAT) AcuA[22] (Fig. 1a). Recently, a general base mechanism enabling acetylation of lysine 549 of AcsA by AcuA has been suggested[23]. Acetylation of AcsA can be reversed by the histone-deacetylase (HDAC)-type AcuC or the sirtuin-type deacetylases SrtN (Fig. 1a)[24,25].

Here we report the serendipitous finding of a stable complex between AcsA and AcuA when Ac-CoA is absent or limited. In this complex, AcuA inhibits AcsA activity by tightly binding to it, while simultaneously channeling the acetyltransferase activity of AcuA specifically toward AcsA, thereby enhancing its specificity. Structural characterization of the AcuA-AcsA complex unravels a so far unrecognized level of complexity in in regulating Ac-CoA biosynthesis from acetate.

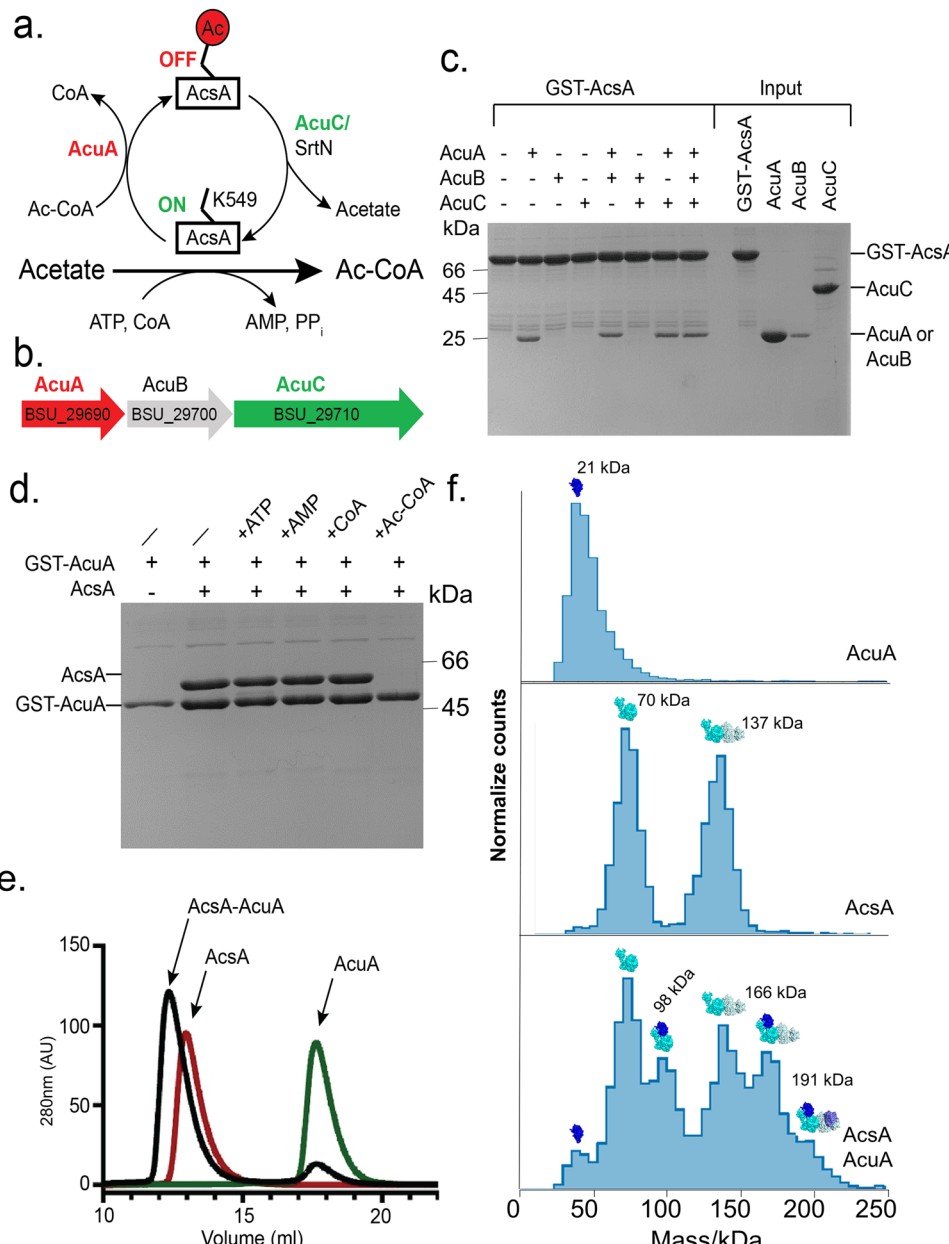

**Fig. 1 | Identification of AcuA-AcsA complexes. a** AcsA activity is regulated by AcuA (red) and AcuC (green) through the Ac-CoA dependent acetylation and deacetylation of K549 (bended stick), respectively. This mechanism tightly regulates capacity of AcsA to convert acetate into Ac-CoA, utilizing ATP and acetate as substrates and producing AMP and pyrophosphate as additional products. **b** Architecture the *acuABC* operon of *B. subtilis* encoding proteins AcuA (red), AcuB (gray) and AcuC (green). The gene locus numbers in *B. subtilis* are also given (compare to: ref. 41). **c** Coomassie-stained SDS-PAGE of a GST-pulldown assay employing GST-AcsA as bait to test interactions with AcuA, AcuB, and AcuC individually as well as in combination. The experiment was repeated three times with similar results. Source data are provided as a Source data file. **d** Coomassie-stained SDS-PAGE of an in vitro pulldown experiment employing GST-AcuA as bait and AcuA as prey to examine the impact of ATP, AMP, CoA and Ac-CoA on the AcsA-AcuA interaction. The experiment was repeated three times with similar results. Source data are provided as a Source data file. **e** Chromatograms of analytical size-exclusion chromatography of AcuA (green), AcsA (red) and the AcsA-AcuA complex. **f** Mass photometry analysis of AcuA (top, 21 kDa), AcsA (middle, monomer 70 kDa, dimer 137 kDa) as well as AcuA-AcsA complexes (bottom, $AcsA_1$-$AcuA_1$, 98 kDa; $AcsA_2$-$AcuA_1$, 166 kDa, and $AcsA_2$-$AcuA_2$,191 kDa).

## Results

### AcsA and AcuA can form a stable complex in the absence of Ac-CoA

AcuA and AcuC are known to acetylate and deacetylate AcsA at K549 to control its activity[8,22]. They are encoded in an operon also coding for the uncharacterized protein AcuB, which shares its promoter region with *acsA* (Fig. 1b)[7]. Originally, we aimed on studying the function of AcuB on AcsA. Thus, we performed in vitro pulldown assays with purified Gluthatione-*S*-transferase (GST)-tagged AcsA as bait and AcuA, AcuB and AcuC as prey proteins. Interestingly, we found a stable interaction between GST-AcsA and AcuA in the absence of Ac-CoA, while no binding of AcuB and AcuC was observed (Fig. 1c). Addition of AcuB, AcuC, or both had no impact on this interaction (Fig. 1c). To verify our data, we used GST-tagged AcuA, which also pulled down AcsA. (Fig. 1d, lanes 1 and 2). Moreover, AcsA-AcuA complex formation in vitro was demonstrated via analytical size-exclusion chromatography (SEC) (Fig. 1e).

AcuA acetylates the catalytic lysine 549 of AcsA to block its activity via this post-translational modification, yielding CoA as a byproduct (Fig. 1a). AcsA activity relies on ATP and CoA as substrates and releases AMP upon formation of Ac-CoA. To test the impact of these compounds (AMP, ATP, CoA, and Ac-CoA) on the interaction we performed in vitro pulldown assays with GST-tagged AcuA as bait (Fig. 1d). Addition of either 1 mM ATP or AMP, or 130 µM CoA did not block the interaction (Fig. 1d, lanes 3, 4 and 5). Upon exposure to 130 µM Ac-CoA the interaction between AcsA and AcuA disappeared (Fig. 1d, lane 6), leading to the acetylation of AcsA proved by immunoblot experiments (Supplementary Fig. 1). Thus, the AcsA-AcuA complex can be disrupted by Ac-CoA, while ATP, AMP as well CoA had no detectable impact under the conditions we tested.

To further characterize the AcuA-AcsA complex, mass photometry (MP) was employed. This technique allows determining the stoichiometric composition of protein complexes at near physiological concentrations (i.e., nanomolar range), which is in contrast to other techniques, such as e.g., SEC analysis. Measuring AcuA (i.e., 37 nM) alone produced a peak at 21 kDa, indicative for a monomer (Fig. 1f, upper panel). MP analysis of AcsA alone (i.e., 37.5 nM) resulted in detection of a monomer (71 kDa) and a dimer (140 kDa) (Fig.1f, middle panel). After combination of 37.5 nM AcsA with 37.5 nM AcuA we observed three additional peaks at 98, 166, and 191 kDa reflecting the following complex compositions: $AcsA_1$-$AcuA_1$, $AcsA_2$-$AcuA_1$, and $AcsA_2$-$AcuA_2$ (Fig. 1f, lower panel). Thus, AcsA-AcuA complexes occur in different compositions with a tetramer consisting of 2 AcsA and 2 AcuA as the maximum.

### Cryo-EM reveals an intertwined interaction interface of AcsA and AcuA

To understand the molecular architecture of the AcsA-AcuA protein complex, we conducted a structural analysis via cryo-electron microscopy (cryo-EM) using a Titan Krios 300 kV electron microscope (FEI) equipped with a K3 camera (Gatan), acquiring a total of 6152 movie frames. Initial three-dimensional (3D) classification ($T = 4$) of 1,194,314 selected particles using AlphaFold-predicted AcsA dimer as starting model yielded two distinct states (Supplementary Fig. 2, Supplementary Table 1). After multiple rounds of processing, those two states were turned out to be AcsA homodimer and AcsA-AcuA complex, determined at 2.89 Å and 2.93 Å resolution, respectively (Supplementary Table 1, Supplementary Figs. 2 and 3).

In the case of AcsA homodimer the NTDs align in a 'back-to-back' orientation with their active sites pointing away from each other (AcsA dimer PDB-ID: 9G79; Fig. 2a, b). This arrangement of the two NTDs in the AcsA homodimer is different from previously reported crystal structures of bacterial and fungal Acs enzymes, which either appeared as monomers like Acs in *Salmonella enterica*[6] or as a trimer in the case of *Saccharomyces cerevisae*[5,26] (Supplementary Fig. 4). Both CTDs present within the AcsA homodimer structure were not resolved

through the large degree of flexibility introduced by a flexible linker connecting the NTDs with their respective CTDs. This is in agreement with mechanistic and structural studies on Acs enzymes in other systems, which demonstrated that flexibility of the CTD is a feature important for the stepwise synthesis of Ac-CoA[5,6].

A second major class of particles showed the AcsA homodimer bound to AcuA (Fig. 2c). In these particles, the CTD of the AcsA subunit of the homodimer bound to AcuA was resolved, while the CTD of the non-AcuA-bound AcsA subunit in the homodimer remained unresolved (Fig. 2c). Our structure clearly demonstrates that AcuA restricts the conformational flexibility of the CTD, thereby enforcing a so far unreported conformation of the NTD and CTD of AcsA.

In the AcsA-AcuA complex (PDB-ID: 9G7F) the CTD of AcsA is bound to the catalytic domain of AcuA, while the very C-terminus of AcuA interacts with the NTD of AcsA (Fig. 2c, d). According to the PISA prediction tool[27], the interface is made up by a combination of polar interactions, salt bridges as well as van der Waals contacts. AcsA and AcuA contribute 1,859.8 Å$^2$ and 1,861.9 Å$^2$ of solvent accessible area, respectively, to the interface of the complex. The perfect complex formation significance score (CSS) of 1.0 corroborates the crucial role of these regions for the complex.

We noticed two significant details in the structure, which might be of functional relevance. First, the very C-terminus of AcuA (amino acids 203 – 210) reaches out into the active site of the AcsA NTD. A detailed analysis revealed that the main chains of AcuA_M209 and Y208 form hydrogen bonds with AcsA_D255, while the main chain of AcuA_Y210 forms a hydrogen bond with AcsA_R108. Additionally, other residues such as the main chains of AcuA_F210, R203, and D200 show weak interactions with AcsA_A303, F283, and R282, respectively (Supplementary Fig. 5). Structural comparison of the NTD of the AcsA-AcuA complex and those of an AcsA-NTD bound to the substrates of the second half reaction, Ac-AMP and CoA, (PDB-ID: 1PG4)[6]; shows that this region of AcuA would overlap with CoA, which may affect AcsA activity (Fig. 2e; see below). Additionally, a loop containing the catalytic residue K549 within the CTD of AcsA aligns with the active site of AcuA (Fig. 2f).

We could confirm the binding interface observed in the cryo-EM structure by a hydrogen-deuterium exchange (HDX) experiment. Hereby, the HDX profile of both proteins present in the AcuA-AcsA complex was compared with that of both individual proteins, and hence changes in HDX reflect conformational changes upon complex formation (Supplementary Figs. 6 and 7). The binding of the AcuA C-terminus into AcsA is confirmed by HDX protection of the AcsA active site-lining residues 253–264 and 301–317, and the AcuA C-terminus (Fig. 2g). Insertion of the AcsA K549-containing loop into the AcuA active center is suggested by HDX protection apparent for both entities (Supplementary Figs. 6 and 7). Elevated HDX apparent for a short stretch of the AcsA FL (residues 459–461) and its CTD (residues 530-538) suggests further conformational changes accompanying formation of the AcuA/AcsA complex.

To validate the relevance of K549 of AcsA and the C-terminus of AcuA for establishing the interface of the complex, we tested the ability for complex formation of an AcsA variant carrying an alanine substitution of K549 and an AcuA variant lacking the last 7 residues among other perturbations (Fig. 2h). Both changes substantially affected the interaction as evident from pull-down assays (Fig. 2h), while the other mutations around the interface had no significant effect on complex formation (Supplementary Fig. 8). Taken together, our data reveal the structure of the AcsA and AcuA – it involves different sites in all domains of the two partner proteins.

### AcuA restricts AcsA activity in an acetyl-CoA-independent manner

To address how the AcsA-AcuA complex alters the activity of AcsA, varying ratios of AcsA to AcuA (1:0.5, 1:1, 1:2, and 1:5) were premixed and then subjected to AcsA activity assays in the presence of 1 mM CoA

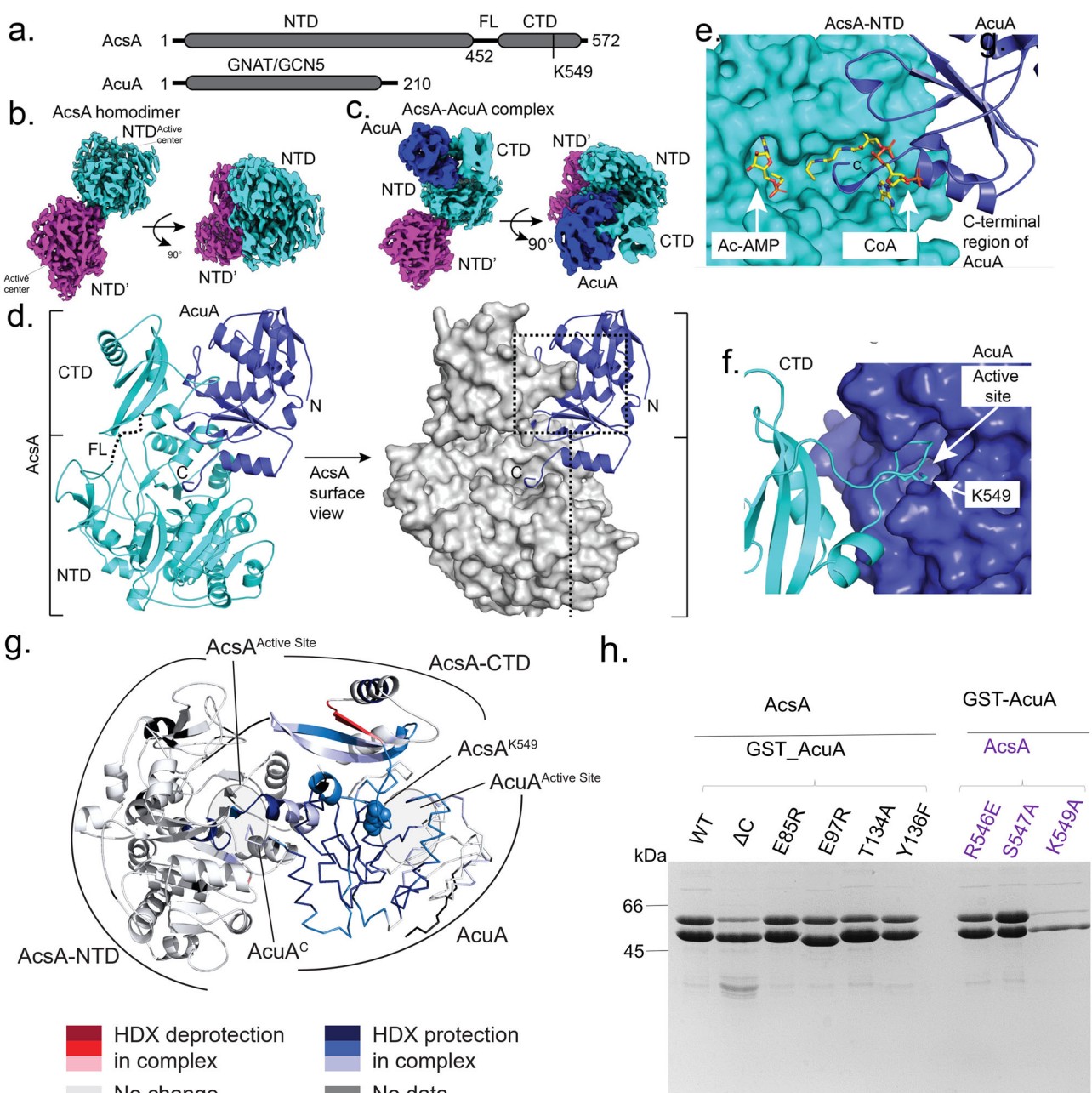

**Fig. 2 | Structural analysis of the AcsA-AcuA complex. a** Domain topology of AcsA (*up*) and AcuA (*down*), drawn to scale. The abbreviations are: NTD: N-terminal domain (orange), CTD: C-terminal domain (red), FL: flexible linker (dotted line), and GNAT/GCN5 protein familiy (blue). **b** Density map of an AcsA homodimer (PDB number: 9g79), two NTD (cyan and magenta) is shown. **c** Density map of AcsA₂-AcuA₁ complex. AcuA (blue) is inserted in AcsA_CTD (cyan) and AcsA_NTD (cyan). **d** Cartoon and surface representation of the cryo-EM structure of the AcsA-AcuA complex (PDB number: 9g7f). **e** AcuA_CTD (blue cartoon) inserted into CoA-

binding site in AcsA (cyan surface). **f** Loop including K549 (cyan cartoon) embedded into active sites of AcuA (blue surface). **g** Changes in HDX occurring in the AcsA-AcuA complex compared to the individual proteins. **h** Coomassie-stained SDS-PAGE showing the results of pulldown experiments of AcsA_WT with GST-AcuA mutants and of AcsA mutants with GST-AcuA_WT conducted in the absence of Ac-CoA. The experiment was repeated three times with similar results. Source data are provided as a Source data file.

and 2.5 mM ATP with different concentrations of acetate ranging from 0.02 to 5 mM. Compared to AcsA without any AcuA, all the mixtures exhibited a significantly reduced maximum velocity ($V_{max}$) of Ac-CoA production (from 2167 to 390 nmol mg⁻¹ min⁻¹) and Michaelis–Menten constant ($K_M$, from 0.43 to 0.10 mM) (Table 1). This shows that AcuA restricts activity of AcsA, probably best described as a "mixed type" inhibitor (Fig. 3a). This data may reflect that AcuA acts as a dual inhibitor – it restricts AcsA activity through the AcuA-AcsA complex ("Ac-CoA independent inhibition") and inactivates AcsA by acetylating the catalytic lysine 549 when sufficient Ac-CoA is present. Therefore, we

needed a workaround to decipher the individual contributions of the two mechanisms to the AcuA-dependent control of AcsA activity. Thus, we constructed the AcuA_E102Q variant, which is incapable of catalyzing acetylation at AcsA_K549[23]. First, we verified that this variant can still form a complex with AcsA via a GST-pulldown experiment (Fig. 3b). Next, we performed an activity assay in which we utilized GST-AcuA and the non-interacting GST-AcuA_ΔC as control proteins. This experiment revealed that AcsA can also be inhibited by binding AcuA_E102Q. It was less potent compared with AcuA but more potent than AcuA_ΔC. To further verify the effects of AcuA_E102Q on AcsA, we

## Table 1 | Kinetic parameters of AcsA activity at varying AcuA concentrations

| AcsA: AcuA | $V_{max}$ (nmol mg$^{-1}$ min$^{-1}$) | $K_M$ (mM) |
|---|---|---|
| ●— wo AcuA | 2167 | 0.43 |
| ⊞— 1:0.5 | 1834 | 0.28 |
| △— 1:1 | 1498 | 0.37 |
| ✛— 1:2 | 908 | 0.23 |
| ✖— 1:5 | 390 | 0.10 |

$V_{max}$ and $K_M$ values were determined using the Michaelis–Menten equation in GraphPad software.

performed kinetic analyses using various AcuA_E102Q to AcsA ratios, similar to those used for AcuA_WT (Supplementary Fig. 9). These results indicate that AcuA_E102Q inhibits AcsA at all tested ratios, and inhibition is stronger when higher AcuA_E102Q concentrations were used. These data again support our hypothesis of two distinct modes of inhibition (Fig. 3c).

The cryo-EM structure of the complex shows that the very C-terminus of AcuA inserts into the CoA-binding site of the NTD of AcsA (Fig. 2e). In the HDX result, we also observed protection of this region upon complex formation (Fig. 3d). We wondered whether the C-terminus of AcuA alone could inhibit AcsA function. Thus, we made use of a peptide representing the C-terminus of AcuA (amino acid

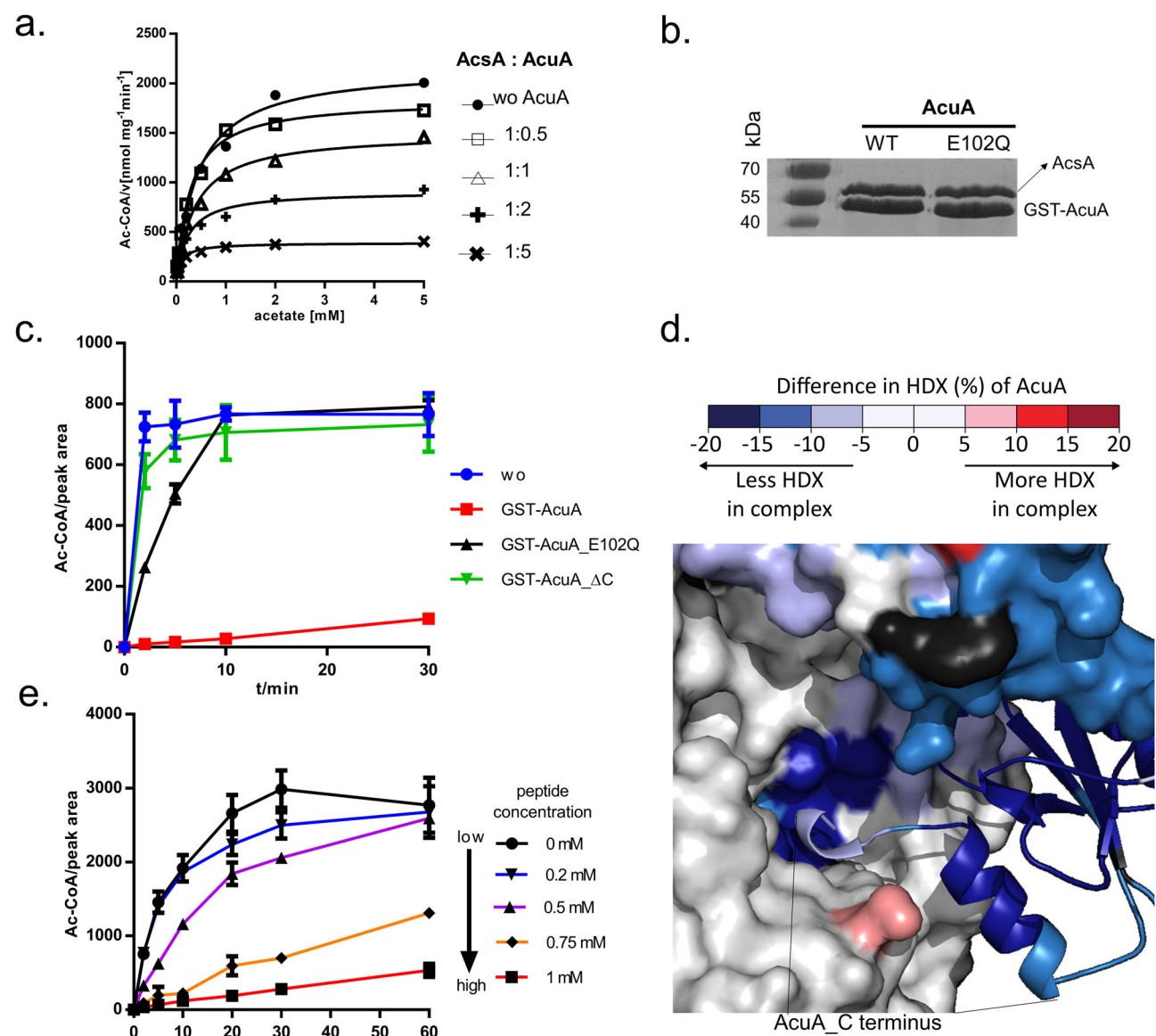

**Fig. 3 | AcuA inhibits the activity of AcsA. a** Kinetic analysis of AcsA activity at varying AcuA concentrations. The data was fitted to a Michaelis–Menten curve using GraphPad software, plotted is the mean of three technical replicates. Source data are provided as a Source Data file. **b** Coomassie-stained SDS-PAGE showing the results a GST-pulldown experiment using GST-AcuA and GST-AcuA_E102Q as bait to assess the interaction with AcsA. Source data are provided as a Source Data file. **c** The effect of GST-AcuA_E102Q on AcsA activity was compared to the effects of AcuA_WT and AcuA_ΔC. The activity assay was also performed without addition

AcuA and error bars represent standard deviation of the mean of three independent experiments. Source data are provided as a Source Data file. **d** Differences in HDX of the AcuA C-terminus (residues 200–210) was analyzed in AcuA-AcsA complexes compared to AcuA alone, across a time range of 10 to 10,000 s. **e** AcsA activity in the presence of different concentrations (0–1 mM) of the AcuA C-terminus peptide (201-RLRFYHRYMY-210). Plotted are the means of three independent experiments. Error bars represent standard deviation of the mean. Source data are provided as a Source Data file.

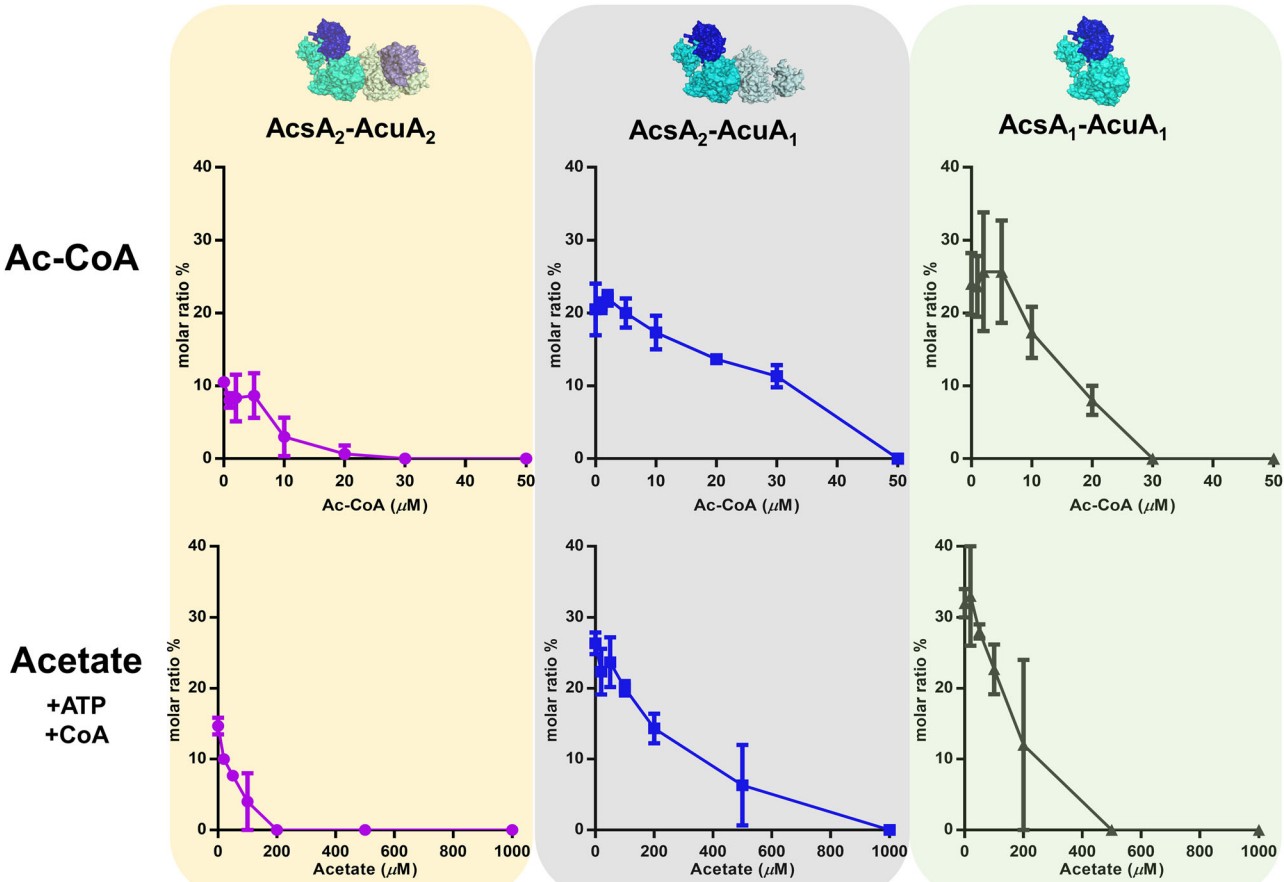

**Fig. 4 | Ac-CoA and acetate disrupt the AcsA-AcuA complexes at highly different concentration.** The effect of varying Ac-CoA concentrations (top) on the formation of the AcuA-AcsA complexes including AcsA$_2$-AcuA$_2$ (*left*), AcsA$_2$-AcuA$_1$ (*middle*), and AcsA$_1$-AcuA$_1$ (*right*), was determined by mass photometry. The effect of varying acetate concentrations on t AcuA-AcsA complexes in the presence of saturating concentrations of CoA and ATP was also determined as described above. The ratio represents the N counts (peak of interest)/N counts (total) and plotted are the means of three independent experiments. Error bars represent standard deviation of the mean. Source data are provided as a Source Data file.

residues: [201]RLRFYHRYMY; Supplementary Fig. 10), and tested its effect on the catalytic activity of AcsA. We found that this peptide interfered with AcsA activity. Addition of the peptide in a final concentration of 1 mM reduced the activity by approximately 90% (Fig. 3e), while lower concentrations were less effective. These experiments corroborate our model that the very C-terminus of AcuA is an important element for the "Ac-CoA independent inhibition" of AcsA.

## AcsA-AcuA complex is sensitive to physiological Ac-CoA concentrations

We could show that AcsA and AcuA form stable complexes in the absence of Ac-CoA. Moreover, AcuA inhibits AcsA by two mechanisms: i. the "Ac-CoA independent inhibition", which relies on the AcsA-AcuA complex and ii. the "post-translational inactivation" involving acetylation of lysine 549 and leading to the disruption of the AcsA-AcuA complex.

Next, we aimed to determine the Ac-CoA concentrations that would disrupt the AcsA-AcuA complex, thereby identifying the concentration thresholds at which each of the two inhibition mechanisms becomes relevant. To this end, we added increasing concentrations of Ac-CoA and analyzed the behavior of the different AcsA-AcuA complexes after 1 min incubation via mass photometry. Increasing concentrations of Ac-CoA did not impact the ratio of the AcsA homodimer to monomer in presence of AcuA but diminished the AcsA$_2$-AcuA$_2$, AcsA$_2$-AcuA$_1$, AcsA-AcuA complexes (Fig. 4). The AcsA$_2$-AcuA$_2$ complex appeared to be most susceptible (Fig. 4, Supplementary Fig. 11). This trend suggests a cooperative mechanism in the Ac-CoA-dependent release of AcuA from AcsA. As a control, we incubated AcsA in isolation with 200 μM Ac-CoA and observed no changes in its oligomerization status (Supplementary Fig. 12). In general, the Ac-CoA concentration required to disrupt the AcsA-AcuA complex was at the lower end of Ac-CoA concentrations reported to occur in bacteria in vivo (i.e., 20–600 μM)[28,29].

We also addressed the influence of acetate on the AcuA-AcsA complexes. We incubated the AcsA, AcuA with 500 μM ATP and 500 μM CoA to simulate cellular concentrations of these compounds. Subsequently, acetate concentrations ranging from 0.02 mM to 5 mM were introduced and allowed to react for 60 min. Following this incubation period, the mixture was subjected to MP analysis. Similar to Ac-CoA, rising concentrations of acetate reduced the formation of AcsA$_2$-AcuA$_2$, AcsA$_2$-AcuA$_1$, AcsA-AcuA complexes, but did not affect the oligomerization of AcsA. This reduction of complex may be caused by production of Ac-CoA in the reaction mixture due to the high acetate levels. Specifically, AcsA$_2$-AcuA$_2$, AcsA$_1$-AcuA$_1$, and AcsA$_2$-AcuA$_1$ complexes were disrupted at acetate concentrations of 200 μM, 500 μM, and 1000 μM, respectively (Fig. 4, Supplementary Fig. 13). These results demonstrate that a millimolar concentration of acetate are required to completely disrupt the AcsA-AcuA complex, subsequently leading to the acetylation of AcsA; in the presence of saturated ATP and CoA in the tested conditions.

In summary, these results reveal an additional layer of regulation contributing to the tightly controlled activation of AcsA. High cellular acetate concentrations seem to be necessary for activating AcsA, while

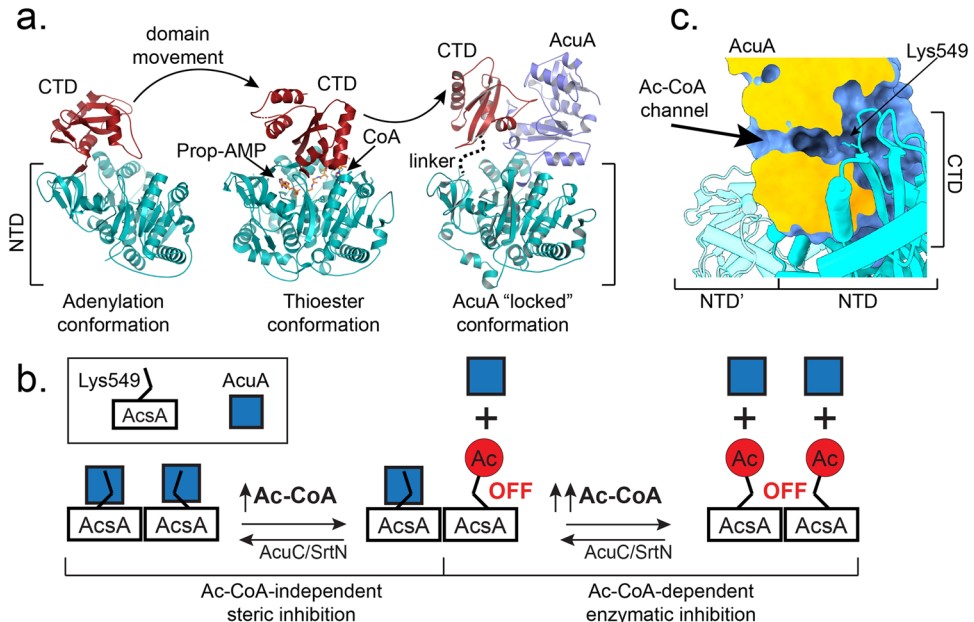

**Fig. 5 | AcuA as a dual inhibitor of AcsA. a** Synthesis of Ac-CoA relies on the rearrangement of the N- and the C-terminal domains of AcsA (NTD in cyan and CTD in red, respectively). Left and middle panels show that adenylate and thioester conformations of AcsA (PDB-IDs: 1T5D and 1PG3, respectively[30]). Right panel: The cryo-EM structure of the AcuA·AcsA complex (this study, PDB-ID: 9G7F) shows that AcuA (blue) "locks" the CTD of AcsA in a position not suitable for catalysis. All structures were superimposed on the NTD of AcsA. **b** AcuA (blue) can restrict the activity of AcsA in two different ways: firstly, a steric mechanism independent of Ac-CoA and, secondly, an enzymatic mechanism involving the Ac-CoA dependent acetylation of Lys549 (red) (compare left to the right side). **c** The cryo-EM structure of the AcuA·AcsA complex (this study, PDB-ID: 9G7F) suggests the presence of a channel allowing the direct access of Ac-CoA into the catalytic side of AcuA (shown as sliced surface) for the direct acetylation of Lys549 within the CTD of AcsA.

in their absence, the complex remains intact, keeping AcsA in its inactive state.

## Discussion

### An unreported mode to sterically restrict AcsA activity in an Ac-CoA independent manner

Acetate, when activated by Ac-CoA synthetases, is an energy expensive source of Ac-CoA. Therefore, this pathway should be effectively paused unless it is absolutely necessary. A key aspect of this regulation is the inactivation through covalent modification of a highly conserved, catalytic lysine residue in the Acs enzymes (i.e., Lys549 in *B. subtilis* AcsA). This modification occurs in an Ac-CoA-dependent manner via acetyltransferases (i.e., AcuA in *B. subtilis*). Although catalytic acetylation of Acs enzymes has been documented in both prokaryotic and eukaryotic systems, a stable complex between an acetyltransferase and an Acs enzyme has never been observed until recently[23]. Our structural data and analysis reveal that AcsA and AcuA from *B. subtilis* form a highly interlinked complex, where each partner obstructs a critical catalytic site of the other. Specifically, the C-terminal domain of AcsA, including the catalytic Lys549, interacts with AcuA's acetyltransferase domain, while AcuA's C-terminus fits into the CoA-binding site in AcsA's N-terminal domain.

These interactions not only create a tight interaction interface between the two proteins, but moreover lock the two domains of AcsA into an arrangement distinct from the catalytic active dimer (Fig. 5a). Ac-CoA synthesis catalyzed by Acs enzymes relies on the dynamic movement and the tight cooperation of its C- and N-terminal domains, executing the first and second step of the reaction[3,5,6,30] (Fig. 5a, *left and middle panels*). The AcsA domain arrangement enforced by the binding of AcuA appears unsuitable for a productive synthesis of Ac-CoA catalyzed by AcsA and may thus be considered as the "locked" state (Fig. 5a, *right panel*). Our biochemical assays indeed show that AcuA acts as an effective inhibitor of AcsA even in the absence of Ac-CoA or its key catalytic residue E102 (Fig. 3c). Therefore, our study extends the

understanding of the regulation of AcsA enzymes from "post-translational inhibition" to an "Ac-CoA-independent steric inhibition" (Fig. 5b). High acetate levels are needed to disrupt the complex, adding to the burden of Acs activation. This occurs because acetate both activates Acs to produce acetyl-CoA, which disrupts the AcuA-AcsA complex, and inhibits it by promoting acetylation of AcsA at K549 via AcuA. In addition, formation of the locked state will reduce futile cycles of protein acetylation and deacetylation if Ac-CoA is limited and, therefore, prevent unnecessary turnover of this valuable compound. Both proteins can, hence, neutralize each other to some extent via "binding-only".

### The AcuA-AcsA complex resembles a preactivated state

The central metabolite Ac-CoA can also function as a second messenger, determining the balance between cellular catabolism and anabolism[31]. Our structural analysis reveal that AcuA bound to AcsA is in an ideal position to execute the post-translational inactivation via acetylation once Ac-CoA is present. In the AcsA-AcuA complex, the C-terminal domain of AcsA sticks its catalytic Lys549 into the active site of AcuA, resembling a pre-catalytic position. However, the step requires an Ac-CoA within the active site of AcuA, raising the question how the acetyl donor would enter the active site of AcuA. Based on our ITC measurements, Ac-CoA exhibits a binding affinity to AcuA with a $K_D$ of $9.47 \pm 0.51\,\mu M$ (Supplementary Fig. 14). We could identify channel within AcuA, which would be suited for entry of Ac-CoA into the active site (Fig. 5c). This idea is also supported by HDX data of AcuA complexed with Ac-CoA, in which the channel region reveals significant protection compared to that of AcuA alone, again on the functional relevance of the channel (Fig. 5b, Supplementary Figs. 15 and 16). In this way, the "Ac-CoA-independent inhibition" AcsA-AcuA complex directly enables inactivation by lysine acetylation.

Interestingly, recent work demonstrated another activity of AcuA: its ability to use acetylphosphate (Ac-P) and CoA to produce Ac-CoA, which can then directly serve as a substrate for acetylation dependent

inactivation of AcsA[15]. AcuA appears to mirror another Ac-CoA producing machine in the cell, the AckA/Pta pathway[32–34] and consequently acts as a sensor for acetylphosphate, a better substrate for Ac-CoA production[15]. Our structural data are in agreement with this model. Thus, in conjunction with these findings and other previous reports[23,35], we propose that AcuA operates through three different modes of inactivation on AcsA. It can sterically inhibit AcsA activity by forming an intertwined neutralization complex reaching out into the active centers of both Acs half reactions (Fig.5a). In addition, AcuA appears to sense Ac-CoA, Ac-P levels and probably even acetate levels, which causes the inactivation and dissociation of the complex.

In addition to the AcsA$_2$-AcuA$_2$ heterotetramer, we observed other species of the complex, including AcsA$_2$-AcuA$_1$ and AcsA$_1$-AcuA$_1$. The experiment was conducted within a nanomolar range, which closely approximates cellular concentrations. With increased Ac-CoA concentrations, we observed that AcsA$_2$-AcuA$_2$ was disrupted first, while AcsA$_2$-AcuA$_1$ required a higher concentration of Ac-CoA to be disrupted (Fig. 4) suggesting that depending on the Ac-CoA levels different sub-complexes with different activities will exist enabling a dynamic control of Acs activity (Fig. 5b). This may be important, since the corresponding genes are strictly co-regulated and formation of different sub-complexes is another way to only keep a subset of Acs enzymes in an active state.

Taken together, the interplay of enzymatic and non-enzymatic inhibition mechanism enables a very tight regulation of this central enzyme. Similar protein complexes including GNATs may form in other biological systems including mammalian cells to integrate metabolite concentration of acetate, AcP, CoA, Ac-CoA, as well as the levels of ATP and AMP into an output such as Acs activity. In mammalian cells, Acs is known to be acetylated, but the enzyme responsible for this acetylation is yet elusive[18,20]. Overall, our study reveals how a key metabolic enzyme and its major regulator assemble to control Ac-CoA production and provides structural insights into the molecular-mechanistic regulation and specificity of protein acetylation .

## Methods

### Production and purification of proteins

Genes encoding for the *Bacillus subtilis* proteins AcuA, AcuB, AcuC and AcsA were amplified from genomic DNA prepared from *B. subtilis* strain 3610. The genes were cloned into pET24d-based plasmid with either N- or C-terminal hexa-histidine tags via a golden gate approach involving BsaI restriction sites. Whenever a GST-tagged protein was required, the same approach was applied.

All proteins and their variants were overexpressed in *Escherichia coli* BL21(DE3) (Novagen) in LB medium supplemented with 10 g per litre D(+)-lactose-monohydrate, 50 µg per mL kanamycin or 100 µg per mL ampicillin (depending on the resistance of vector) for 16–20 h at 30 °C and rigorous shaking. After centrifugation at 4500 × *g*, 20 min, 4 °C, the obtained cells were resuspended in buffer A (20 mM HEPES-Na pH 8.0, 250 mM NaCl, 20 mM KCl, 20 mM MgCl$_2$ and 40 mM imidazole) and were lysed through an LM10 microfluidizer (Microfluidics) at 15,000 psi. Lysate was then centrifuged 125,000 × *g*, 20 min, 4 °C. Supernatant was loaded onto a HisTrap HP 1 mL column (Cytiva) equilibrated with buffer A. 10 mL buffer A was used for washing step and subsequently eluted by 10 mL buffer B (20 mM HEPES-Na pH 8.0, 250 mM NaCl, 20 mM KCl, 20 mM MgCl$_2$ and 250 mM imidazole). The obtained elution was concentrated via Amicon Ultracel-30K (Millipore) to 2 mL and applied to SEC (HiLoad 26/600 Superdex 200 pg, Cytiva) equilibrated with SEC buffer (20 mM HEPES-Na pH 7.5, 200 mM NaCl, 20 mM KCl and 20 mM MgCl$_2$). The fractions containing target proteins were then concentrated via Amicon Ultracel-30K (Millipore) and snap-frozen in liquid nitrogen and stored at -80 °C for the subsequently analysis. Protein concentrations were determined photometrically with a NanoDrop Lite (Thermo Fisher).

### In vitro pulldown assays

The GST pulldown assays were performed using spin columns and filters purchased from MobiTec. A 20 µL suspension of GST-Sepharose beads from GE Healthcare was loaded into a spin column and resuspended in 450 µL of SEC buffer. After centrifugation for 1 min at 4200 × *g*, 200 µg of GST-tagged protein was immobilized on the beads by incubating on a rotation machine for 15 min, followed by centrifugation under the same conditions. The immobilized protein was washed with 500 µL of SEC buffer and centrifuged. Next, 200 µg of the interaction partner was loaded onto the column in 400 µL of SEC buffer and incubated with the GST-tagged protein on a rotation machine for 30 min. After three washes with 500 µL of SEC buffer, the bound proteins were eluted with 50 µL of SEC buffer supplemented with 20 mM GSH (pH 8.0). All samples were then separated by SDS-PAGE and stained with Coomassie blue for analysis.

### Cryo-electron microscopy (cryo-EM)

Following a 30-min incubation at room temperature, AcuA was combined with AcsA in a 3:1 ratio. The resultant mixture was then injected into a Superdex 200 HiLoad 16/600 column. Fractions containing the AcsA-AcuA complex were pooled, concentrated, and subsequently stored at -80 °C.

For cryo-EM sample preparation, UltrAuFoil R1.2/1.3 300 mesh gold grids were employed. A volume of 3.5 µL containing the AcsA-AcuA complex at a concentration of 0.5 mg/mL was applied onto the grid within a Vitrobot Mark IV chamber (FEI), maintained at 4 °C with 100% humidity. The grid underwent blotting with two layers of filter papers for 4 s at a force level of 0. Subsequently, the grid was subjected to imaging using a Titan Krios 300 kV electron microscope (FEI) equipped with a K3 camera (Gatan). Imaging was performed at a magnification of ×105,000 with pixel sizes set at 0.708 Å/pixel, utilizing EPU in super-resolution mode (binning 2) and faster acquisition mode. The defocus range was set between −1.0 µm to −2.2 µm, with a total electron dose of 50 electrons per square angstrom (e⁻/Å$^2$).

Data processing was performed using RELION[35]. Initially, movies underwent motion correction with dose-weighting and gain correction. Subsequently, the resulting micrographs were subjected to contrast transfer function (CTF) estimation using CTFFIND4.1. Templates for autopicking were generated based on approximately 3000 manually picked particles. A total of 2,969,170 auto-picked particles were downsized by a factor of 2 and subjected to 2D classification with the option to ignore CTFs enabled. Following this, 1,194,314 particles from high-quality 2D classes were selected for 3D classification using Alphafold 2-predicted AcsA dimer as the initial model, with T values of 4 or 2.

Two states were identified from the initial 3D classification, representing AcsA-AcuA complex and AcsA homodimer, respectively. However, the AcsA dimer appeared noisy, so we performed a parallel 3D classification using a lower T value (T = 2). From the 3D classification, 3 classes from T = 4 and 1 class from T = 2 representing the AcsA-AcuA complex, as well as 2 classes from T = 2 representing the AcsA dimer, were chosen for further processing. This involved re-extraction, 3D classification/refinement with or without blush, particle polishing, CTF refinement, particle reweighting, postprocessing, and local resolution estimation. Ultimately, the structures of the AcsA-AcuA complex and the AcsA dimer were resolved to resolutions of 2.93 Å and 2.89 Å, respectively (Supplementary Table 1).

### Hydrogen/deuterium exchange mass spectrometry (HDX-MS).

HDX-MS experiments were conducted essentially following previously established protocols[36]. In experiment 1 probing AcuA/AcsA complex formation, individual proteins or the complex obtained by SEC were employed at 50 µM stock concentrations. In experiment 2 probing acetyl-CoA-binding to AcuA, protein and ligand were employed at stock concentrations of 50 µM and 10 mM, respectively (Supplementary Table 2). HDX reactions were prepared by a two-arm robotic

autosampler (LEAP technologies). 7.5 µL of sample were pre-dispensed and then 67.5 µL of HDX buffer (20 mM HEPES-Na pH 7.5, 20 mM KCl, 20 mM MgCl₂, and 200 mM NaCl) prepared with 99.9% D₂O added. After incubation at 25 °C for 10, 30, 100, 1000 or 10,000 seconds, 55 µL of the HDX reaction were withdrawn and added to 55 µL of pre-dispensed quench buffer (400 mM KH₂PO₄/H₃PO₄, pH 2.2, 2 M guanidine-HCl) kept at 1 °C. 95 µL of the resulting mixture were injected into an ACQUITY UPLC M-Class System with HDX Technology (Waters)[37]. Preparation of non-deuterated samples was conducted similarly (incubation for approximately 10 s at 25 °C) by tenfold dilution of samples with HDX buffer prepared with H₂O. Injected samples were flushed out of the 50 µL-loop with H₂O + 0.1% (v/v) formic acid (100 µL/min), guided to a column (2 mm × 2 cm, 12 °C) containing porcine pepsin immobilized to beads, and the resulting peptic peptides were trapped on an ACQUITY UPLC BEH C18 1.7 µm 2.1 × 5 mm VanGuard Pre-column (Waters) kept at 0.5 °C. After 3 min of digestion and peptide trapping, the trap column was placed in line with an ACQUITY UPLC BEH C18 1.7 µm 1.0 × 100 mm column (Waters), and the peptides eluted at 0.5 °C using a gradient of eluents A (H₂O + 0.1% (v/v) formic acid) and B (acetonitrile + 0.1% (v/v) formic acid) at a flow rate of 30 µL/min as follows: 0-7 min: 95–65% A; 7–8 min: 65–15% A; 8–10 min: 15% A; 10–11 min: 5% A; 11–16 min: 95% A. Eluting peptides were guided to a G2-Si HDMS mass spectrometer with ion mobility separation (Waters) and ionized with an electrospray ionization source (250 °C capillary temperature, 3.0 kV spray voltage) and mass spectra acquired in positive ion mode over a range of 50 to 2000 $m/z$ in enhanced high definition MS (HDMS$^E$) or high definition MS (HDMS) mode for non-deuterated and deuterated samples, respectively[38,39]. Lock-mass correction was implemented with [Glu1]-Fibrinopeptide B standard (Waters). During separation of the peptide mixtures on the ACQUITY UPLC BEH C18 column, the protease column was washed three times with 80 µL of wash solution (0.5 M guanidine hydrochloride in 4% (v/v) acetonitrile), and blank injections performed between each sample to reduce peptide carry-over. The experiments were conducted on three individually purified protein batches whereby for each protein state and time point three technical replicates (individual HDX reactions) were measured.

Peptide identification and analysis of deuterium incorporation were carried out with ProteinLynx Global SERVER (PLGS, Waters) and DynamX 3.0 softwares (Waters)[36]. In brief, peptides were identified with PLGS from the non-deuterated samples acquired with HDMS$^E$ by employing low energy, elevated energy, and intensity thresholds of 300, 100 and 1,000 counts, respectively. Identified ions were matched to peptides with a database containing the amino acid sequence of AcuA, AcsA, porcine pepsin, and their reversed sequences with the following search parameters: peptide tolerance = automatic; fragment tolerance = automatic; min fragment ion matches per peptide = 1; min fragment ion matches per protein = 7; min peptide matches per protein = 3; maximum hits to return = 20; maximum protein mass = 250,000; primary digest reagent = non-specific; missed cleavages = 0; false discovery rate = 100. Only peptides that were identified in three non-deuterated samples and with a minimum intensity of 10,000 counts, a peptide length of 5−40 residues, a minimum number of two products, a maximum mass error of 25 ppm and retention time tolerance of 0.5 min were considered for further analysis. Deuterium incorporation into peptides was quantified with DynamX 3.0 software (Waters). All spectra were manually inspected and, if necessary, peptides omitted (e.g., in case of low signal-to-noise ratio or presence of overlapping peptides). Whenever possible, multiple charge states were employed for quantification of deuterium uptake. The observable maximal deuterium uptake of a peptide (Supplementary Table 2) was calculated by the number of amino acid residues minus one (for the N-terminal residue) minus the number of proline residues. For the calculation of HDX in percent the absolute HDX was divided by the theoretical maximal deuterium uptake multiplied by 100. To render the residue-specific HDX differences from overlapping peptides, the

shortest peptide covering a residue was employed. Where multiple peptides were of the shortest length, the peptide with the residue closest to the peptide's C-terminus was utilized.

## Activity assay

The AcsA activity assay was conducted in 50 µL reaction volumes containing 1 mM CoA, 2 mM ATP, 0.1 mM DTT, and 0.2 µg of AcsA. Acetate concentrations varied from 0.02 to 5 mM in the reaction mixtures. Additionally, different quantities of AcuA or AcuA_E102Q were included to examine its impact on AcsA activity. Reactions were allowed to proceed at room temperature for 10 min. An equal volume of acetonitrile was then added, followed by centrifugation at 16,200 × $g$ to remove precipitates. The resulting supernatant was analyzed using High-Performance Liquid Chromatography (HPLC) on an Agilent 1200 series system equipped with a Metrosep A Supp 5−150/4.0 column (Metrohm). Isocratic elution was performed with 90 mM (NH₄)₂CO₃ pH 9.25 at 0.6 mL/min flow rate for 20 min. Detection was carried out at a wavelength of 260 nm.

To test the inhibitory activity of the C-terminal peptides of AcuA on AcsA, a similar method to that used for AcsA kinetics was employed. Specifically, 0.2 to 1 mM of peptides were pre-incubated with 0.2 µg of AcsA for 10 min. Subsequently, the reaction mixture was supplemented with 0.5 mM CoA, 1 mM ATP, 0.1 mM DTT, and 5 mM potassium acetate for the assays. The reactions were terminated at 1, 2, 5, 10, 30, and 60 min for subsequent analysis by using the HPLC system as described above.

To assess the impact of AcuA_ΔC on AcsA activity, 6 µg of GST_AcuA_ΔC was pre-incubated with AcsA for 10 min at room temperature. Following this, the reaction mixture was supplemented with 0.5 mM CoA, 1 mM ATP, 0.1 mM DTT, and 5 mM KAc for the assays. Controls included GST_AcuA_WT and AcsA alone. Reactions were terminated at 2, 5, 10, and 30 min and subsequently analyzed using the HPLC system as described above. All activity assays were repeated three times and fitted to the Michaelis−Menten or other curve.

## Mass photometry

Mass photometry (MP) measurements were recorded on a TwoMP mass photometer (Refeyn Ltd., UK), operated with the AcquireMP v2023 R1.1 software (Refeyn Ltd., UK). Microscope coverslips (1.5 H, 24 × 60 mm, Carl Roth) and Culture Well Reusable Gaskets (CW-50R-1.0, 3 × 1 mm; Grace Bio-labs, USA) were cleaned three times with alternating rinsing steps of isopropanol and ultrapure-grade water (Milli-Q, Merck, Germany), air-dried under a compressed air stream, assembled, and mounted on the mass photometer with immersion oil (Immersol™ 519 F; Carl Zeiss, Germany). For one MP measurement, a drop of 18.5 µL of SEC buffer was applied to one well of the gasket, focused using the "Droplet dilution" option and mixed with 1.5 µL of pre-diluted protein or protein mixture (see below) to measure at a final protein concentration of 37.5 nM in the drop. The mass photometer was calibrated with a custom molecular weight standard with proteins of known sizes (86 kDa to 344 kDa). MP measurements were recorded for 60 s at 100 frames per second. The data was analyzed with the DiscoverMP v2023 R.1.2 software (Refeyn Ltd., UK). Peaks were selected manually and fitted with a Gaussian fit. The total number of counts that were predicted by the Gaussian model and percentage values compared to the total count of binding events were extracted and used to quantify and compare the oligomeric states of interest.

## Western blot assays

To prepare acetylated AcsA (AcsA_K549$^{Ac}$), 10 mg of AcsA, 10 mg of GST-AcuA, and 2 mg of Ac-CoA were incubated in 10 mL of SEC buffer at 37 °C for 60 min. The mixture was then diluted to 40 mL with SEC buffer and passed through a 5 mL GST column to remove the GST-AcuA. The flow-through, containing AcsA_K549$^{Ac}$, was collected and

concentrated to a final concentration of 6.5 mg/mL for subsequent western blot assays.

For the western blot, 8 µL of 0.5 mg/mL AcsA_K549[Ac] and AcsA samples were subjected to SDS-PAGE and then transferred to a PVDF membrane using the BIO-RAD Trans-Blot Turbo system for 7 min at 1.3 A, 25 V. The membrane was then blocked at room temperature for 1 h with 10% (w/v) non-fat dry milk (NFDM) in 1× TBST (20 mM Tris-HCl, pH 7.5, 150 mM NaCl, and 0.1% (v/v) Tween-20). Anti-acetyl-lysine antibodies from rabbit (Merk: SAB5600275) were diluted 1:1500 in TBST/5% (w/v) NFDM and incubated overnight at 4 °C. The blot was washed three times with TBST. For the secondary antibody step, anti-rabbit IgG-alkaline phosphatase antibodies (Cell Signaling Technology, 7074 s) were diluted 1:1500 in TBST/0.5% (w/v) NFDM and incubated at 4 °C for 1 h. Signals were detected using the ECL prime system (ECL prime luminol enhancer solution and ECL prime peroxide solution) and documented with a Fusion-SL chemiluminescent imager (Peqlab).

### Analytical size-exclusion chromatography (SEC)
Purified AcuA and AcsA were prepared at 100 µM in SEC buffer and incubated for 10 min at room temperature. Subsequently, 100 µL of the mixture were injected at 10 °C onto a pre-equilibrated S200 300/10 GL analytical size-exclusion column (GE Healthcare, Munich, Germany) using an Akta system (UNICORN 7.6; Cytiva). The data were plotted using GraphPad Prism (GraphPad Prism Corp., San Diego). Single proteins were also injected and processed using the same procedure.

**Isothermal titration calorimetry.** Ligands and proteins were always diluted in the very same SEC buffer. To measure the interaction between AcuA and acetyl-CoA, the purified protein was titrated in the sample cell at a final concentration of 81–83 µM. The protein concentrations were pre-determined by their absorbance at 280 nm. The ligand was placed in the titration syringe at a nominal concentration of 1 mM to saturate the protein sample during the titrations. The measurements were performed at 25 °C with the instrument MicroCal PEAQ-ITC (©Malvern Panalytical) with a method consisting of 13 injections (first 0.4 µL, and the rest 3 µL each) and 150 s of spacing. The raw data were processed with the MicroCal PEAQ-ITC Analysis Software using the "one binding site" model and plotted using GraphPad Prism.

### Reporting summary
Further information on research design is available in the Nature Portfolio Reporting Summary linked to this article.

## Data availability
HDX-MS data have been deposited to the ProteomeXchange Consortium via the PRIDE[40]. Partner repository with the dataset identifier PXD058390 and the HDX-MS data table is presented in Supplementary Table 2. The Cryo-EM data supporting the findings of this study have been deposited at the PDB with deposition IDs 9G79 and 9G7F and at the EMDB EMD-51121 and EMD-51116. PDB codes of previously published structures used in this study 1T5D, 1PG3 and 1PG4 were downloaded from PDB database. Source Data are provided as a Source data file. Source data are provided with this paper.

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

## Acknowledgements

We would like to thank the Max-Planck-Society for support (to G.B. and G.K.A.H.). L.Z. would like to thank Pietro Giammarinaro for support at the beginning of the project. We acknowledge support by the German Research Council (DFG) through the core facility for HDX-MS (project 324652314 to G.B.). Y.D. thanks the LMB cryo-EM facilities for their support during data collection; J. Grimmett and T. Darling for computing; Chao Qi and Sjors Scheres for suggestions on data processing. Y.D. was supported by the UK Medical Research Council MC_U105184332 and a Wellcome Trust Senior Investigator award (WT096570) to Venki Ramakrishnan, the China Postdoctoral Science Foundation (PC2021083) and the Leverhulme Trust (ECF-2022-525). M.G. acknowledges support by the Peter und Traudl Engelhorn Foundation.

## Author contributions

L.Z., J.F., and G.B. conceptualized and supervised the project, guided the experimental process, and wrote the original draft. L.Z., P.B. and C.N.M. generated constructs, purified fusion proteins and conducted biochemical assays. Y.D. conducted the cryo-EM experiments and determined the complex structure. W.S. performed the HDX-MS experiments. E.J.K. carried out the ITC measurements. M.G. and G.K.A.H. conducted the MP measurements, contributed methodology and supervision. I.B.D. contributed expert advice. F.A. synthesized the AcuA peptide. All authors were involved in data analysis and revising the manuscript.

## Funding

## Competing interests

The authors declare no competing interests
