## [Transparent Peer Review file · Nature Communications]

Regulation of acetyl-CoA biosynthesis via an intertwined acetyl-CoA synthetase/acetyltransferase complex.

Corresponding Author: Professor Gert Bange

Version 0:

Reviewer comments:

Reviewer #1

(Remarks to the Author)

In this study, Zheng, Bange and coworkers elucidate a novel inhibition mode of acetyl-CoA synthetase in *B. subtilis* bacteria. This inhibition occurs through the binding of an acetyl-transferase enzyme (AcuA), encoded in the same operon as AcsA, to AcsA. The authors investigate the molecular interaction between AcsA and AcuA using different techniques, including single particle Cryo-EM, Mass Photometry, Hydrogen-Deuterium Exchange Mass Spectrometry (HDXMS), site-directed mutagenesis and in vitro pulldown assays. They determine the structures of two complexes, the Acs dimer (Acs2) and an Acs dimer bound to one AcuA (Acs2-AcuA1). The cryo-EM dataset yielded 4 maps including the Acs2 dimer, the Acs2-AcuA1 trimer and 2 local refinement maps from the Acs2-AcuA1 complex of the CTD of Acs and the other of the AcuA complex. The C-terminus of AcuA binds near the Acs2 acetyl-CoA binding site, while AcsA K549 (the acetylation of which also inhibits AcsA activity) is found bound to AcuA near its active site. The HDX-MS data corroborates the binding interface found in the cryo-EM structure of Acs2-AcuA1 while the mass photometry data and activity assays show a acetyl-Coa dependent change in the oligomeric state of this complex.

Together, this study reveals a new acetyl-CoA independent mode of AcsA inhibition as an alternative to the acetyl-CoA dependent mode of AcuA-mediated acetylation of lysine 549 on AcsA to mediate AcsA inhibition. The studies are rigorous, well described and illustrated and nicely complements the recently published studies of Qin et al. (ref 23) that comes to similar conclusions using biochemical studies and AlphaFold models. The studies have important implication for understanding the molecular mechanisms of the regulation of acetyl-CoA metabolism in *B. subtilis* with possible implications in other species.

The authors should address the following points prior to publication:

Major points

1. Can the authors say anything about the significance of the structure of the AcsA dimer. Is there something interesting about the interface that provides functional relevance for why the protein forms a dimer?
2. Page 4 – lines 29-30, the authors refer to “the other mutations” that have no effect on the AcsA-AcuA interaction. Can the authors say something about (and illustrate) what these residues are doing.
3. Figure 3a; The mixed type of inhibition (steric vs lysine 549 acetylation) that is observed in this figure would be easier to deconvolute if the authors titrated in a catalytic mutant of AcuA instead of the wildtype protein.
4. Figure 3b; while I assume that the AcuA_deltaC construct is unable to form a complex with AcsA it would be reassuring to confirm this using MP.
5. Figure 4; the authors should show the effect of Ac-CoA addition on AcsA dimer as a control.
6. The physiological connection to acetate concentration described in the discussion is difficult to follow. The authors state “High levels of acetate are required to break up this complex demonstrating that it significantly adds to the burden of Acs activation.” But high levels of acetate would both activate Acs to produce more acetyl-CoA (and disrupt the AcuA complex) and also inhibit it by producing more acetyl-CoA product to acetylate Acs at K549 (via AcuA).

7. Were the authors able to generate a 3D class of the particles with the AcsA2-AcuA2 tetramer? It might be worthwhile to share an image of the other classes from the 3D classification step in Supplementary Fig. s2. The authors could try re-picking particles based on an AlphaFold model of the tetramer complex to see if AcsA2-AcuA2 inhibits AcsA in the same way. Alternatively, the authors could use the Mass Photometry data to optimize the correct substrate and product concentrations to enrich the population of particles pertaining to specific oligomeric states.

Minor points

8. Pg 3, lines 22-24; Use of T values in the results section is too technical for the non-specialist. If the authors want to use this in the results section, they need to define the significant and difference between T2 and T4.

10. Supplemental Fig. 4; the color coding is confusing. To be consistent with the text, it would make the most sense to just color-code the N- and C- term domains.

11. Throughout the text the authors use the term “properly resolved” but it is unclear to me why the word “properly” is used. It is resolved or not resolved.

12. Page 3, line 27: “homdimer” should be ‘homodimer’

13. Page 3, line 30: “PD-ID” should be “PDB ID”

14. Page 3 line 43; change “unknown conformation” to “unreported conformation”

15. Page 3, line 45: “AscA” should be “AcsA”

16. Page 4, line 8-9: “AcuA_F210” should be “AcuA_Y210”

17. Page 6, line 15-16; The sentence that reads “Acetate activated by Ac-CoA synthetases is probably the poorest source for Ac-CoA and thus this pathway needs to be efficiently paused if not absolutely required.” Is confusing. Can the authors reword or elaborate.

Reviewer #2

(Remarks to the Author)

Reviewer #3

(Remarks to the Author)

Overview: The manuscript titled “Dual Inhibition through an Intertwined Acetyl-CoA Synthetase/Acetyltransferase Complex” explores the acetyl-CoA-independent inhibition of acetyl-CoA synthetase (AcsA) by acetyl-CoA acetyltransferase (AcuA), in addition to the known acetyl-CoA-dependent inhibition mechanism. This study integrates biochemical kinetics assays, Cryo-EM structural determination, and HDX-MS analysis to identify the binding interface and elucidate the complex regulatory interactions between AcsA and AcuA as basis for acetyl-CoA-independent inhibition. Much of the data are convincing and the discussion of possible regulatory mechanism is very interesting. However, a number of issues need to be addressed.

I- Lack of quantitative evidence for the importance of acetyl-CoA-independent inhibition:

The manuscript could benefit from providing clearer evidence for acetyl-CoA-independent inhibition of acetyl-CoA synthetase. Based on the cryo-EM modeled complex of AcsA2, the paper shows an interaction between the C-terminal region of AcuA and the active site at the N-terminus of AcsA. Deletion of the C-terminal region (AcuA- Δ C) weakens the interaction between AcuA and AcsA but does not completely abolish complex formation, as seen in the GST-pulldown assay (Fig. 2h).

AcuA exhibits mixed inhibition of AcsA kinetics in the presence of acetyl-CoA-dependent inhibition (Fig. 3A). To establish the role of above interaction in the acetyl-CoA-independent inhibition of AcsA by AcuA, the paper demonstrates that deletion of the Δ C region changes the rate of product formation (Fig 3C). With this information, it was concluded that deletion of the C-terminus of AcuA weakens complex formation and hence may require higher amounts of protein to achieve the same level of inhibition.

Critique: In this context, it would have been very helpful to see quantitative inhibition kinetics data (impact on k_{cat} and K_m) conducted with varying levels of AcuA- Δ C, for comparison to the AcsA + AcuA conditions.

II-HDX-MS Data in the Manuscript: The HDX work presented in this manuscript demonstrates high sequence coverage and good repeatability in data collection. The authors utilized the same set of redundant peptides for both the apo-form and

protein complexes, allowing for direct comparisons between different states. The peptide data from various samples are well-organized in structured tables, made available in PRIDE, which facilitates comparisons across experimental conditions. Additionally, the HDX protection analysis is tabulated and presented.

Critique: There are several points in the manuscript that require further clarification.

i) Lack of Binding Affinity Information (Kd): The HDX-MS experiments were conducted to identify the binding interface between homodimer of AcsA and AcuA (AcsA2:AcuA complex) that was identified in cryo-EM. The HDX-MS of AcsA:AcuA complex were conducted using equimolar concentrations (5 μ M: 5 μ M).

The manuscript does not provide key binding affinity (Kd) values for the various interactions observed. As a result, without information on the binding strengths and relative populations of the different species (AcsA monomer, AcuA monomer, AcsA dimer, AcuA dimer, AcsA:AcuA AcsA2:AcuA and others), it is difficult to interpret the HDX traces with regard to the impact of the alternate protein. The fact that the protection offered by the binding partners disappears at the longer time points indicates significant amount of uncomplexed partners in most instances: protection can be detected at early times but disappears as the protein being assayed undergoes competing, unprotected exchange. The implication of relatively weak binding between AcsA and AcsU would be resolved with the availability of Kd values. Can the authors estimate Kd values from their kinetic assays (Figure 3)?

ii) AcuA-Acetyl-CoA Complexation Studies: It would also be helpful to include key binding affinity (Kd) values to better understand the relative populations of complexed and apo forms in the AcuA-Acetyl-CoA complexation studies.

iii) Lack of Biological Replicates: It appears that the HDX-MS data were obtained using technical triplicates, but no biological replicates are included. Incorporating biological replicates would strengthen the reliability and robustness of the conclusions, as biological variability, which can influence protein behaviour under different conditions, has not been accounted for.

iv) No Peptide-Specific Back-Exchange Correction: Performing peptide-specific back-exchange correction would help ensure the strength of the exchange measurements reported, especially when varying numbers of redundant peptides are used, as is the case in this work.

v) Contradiction Between HDX and Cryo-EM Data: The HDX analysis (summarized in Fig. 2E) shows that the C-terminal domain (CTD) of AcsA is less protected in the complex, implying that the CTD becomes unstructured upon complex formation. However, This observation appears to differ from the cryo-EM data, which shows that the CTD of AcsA becomes resolvable in the AcuA-bound complex, whereas it remains unresolved in the AcsA homodimer (page 3-line 39; Fig 2B, and 2C). More discussion is needed to reconcile this discrepancy.

vi) No HDX Changes Observed in Key Interaction Regions: The acetyl-CoA-independent inhibition proposed in this work is based on the interaction between the C-terminus of AcuA and the active site residues of the N-terminal domain (NTD) of AcsA. However, the HDX data for AcuA (Fig. S7C, peptide 203-210) show no significant protection factor changes upon complex formation. Similarly, the residues in the very end of the NTD of AcsA that interact with AcuA do not exhibit protection changes in the HDX data (Fig. S6C). These findings warrant further clarification to strengthen the proposed mechanism and its role in acetyl-CoA-independent inhibition.

Reviewer #4

(Remarks to the Author)

Reviewer #5

(Remarks to the Author)

The manuscript by Zheng et al., reports the characterization of the structure of a complex between acetyl-CoA synthetase (Acs) and a GCN5-related N-acetyltransferase (AcuA), and the effect of complex-formation on Acs catalytic activity. Acs catalyzes the formation of acetyl-CoA from acetate and CoA, a reaction that requires the hydrolysis of ATP to AMP plus PPI. The Acs-catalyzed reaction involves a catalytic Lysine residue, and in the case of the *Bacillus subtilis* Acs this is lysine residue-549. Acs enzymes from almost all biological sources can be inactivated by the acetylation of this catalytic lysine residue, and in *B. subtilis* this acetylation reaction is catalyzed by AcuA. The authors find that the formation of a Acs-AcuA complex is inhibitory to Acs catalysis, and this complex occurs in the absence of acetyl-CoA - thus they propose that this is an acetyl-CoA-independent mechanism for regulating Acs activity. Thus, two mechanism for regulating acetyl-CoA generation from acetate, one that is dependent on acetyl-CoA and another that is acetyl-CoA independent provides more stringent control.

Suggests:

1) Strictly speaking the acetylation of the active site Lys residue is a "post-translational inactivation" of Acs, and I suggest that the authors use this phraseology rather than "inhibition of Acs". One can thereby better distinguish the Acs-AcuA

complex formation as an inhibitory mechanism from the “post-translational inactivation” by acetylation.

2) However, enzymologically could not the formation of the AcuA-Acs complex be considered as the enzyme-substrate intermediate complex in the acetylation of Acs by AcuA reaction? With Acs being one of the substrates and AcuA being the enzyme. Subsequently the AcuA-Acs intermediate requires the second substrate of the acetylation reaction, namely acetyl-CoA. I suppose one could argue that the novelty is the stable formation of the enzyme-substrate complex formation (AcuA-Acs) in the absence of the second substrate. Then, the question becomes, is this physiologically relevant in vivo and/or is this unique to the acetylation of Acs, or does it occur commonly in other protein acetylation reactions – for example, histone acetylation.

3) Somewhat related to the above, on p5, lines 8-18, the authors find that the C-terminal 7-residues of AcuA mediates the inhibition of Acs by binding to the CoA-binding site of Acs. But I notice that in the experiments shown in Figure 3c, the synthetic C-terminal 7-residue peptide of AcuA that causes 90% inhibition of Acs activity is used at a 1mM concentration. This seems rather high concentration. Is this a physiological concentration? What is the concentration of Acs in these experiments?

Minor points:

a) Figure 4 - The y-axis is labeled as “Molar ratio”, and this number is presented as a percentage. But it’s unclear what ratio was measured? Acs:AcuA? What is 100%? Please better define?

b) The first sentence of the Discussion, the authors state, “Acetate activated by Ac-CoA synthetases is probably the poorest source for Ac-CoA and thus this pathway needs to be efficiently paused if not absolutely required.” It’s not clear what “poorest” means. Could they better define - maybe they are referring to the most “energy expensive” reaction, as the Acs reaction costs 2 molecules of ATP, as ATP is hydrolyzed to AMP, and the reaction needs to be pulled to the acetyl-CoA product side, by the hydrolysis of the other product, pyrophosphate to phosphate.

Version 1:

Reviewer comments:

Reviewer #1

(Remarks to the Author)

The authors did a nice job addressing our concerns, but one important point still remains about the significance of the acetyl-CoA independent inhibition of AcsA by AcuA. This is relevant to the following comments that we made below:

Figure 3a; The mixed type of inhibition (steric vs lysine 549 acetylation) that is observed in this figure would be easier to deconvolute if the authors titrated in a catalytic mutant of AcuA instead of the wildtype protein.

Related to our comment, Reviewer 5 (and also a related point of review 3) bring up a similar concern, noting the possibility that the acetyl-CoA independent inhibition might be an enzyme intermediate as opposed to an independent mode of inhibition.

To address this concern, the authors prepared a AcuA_E102Q catalytic mutant, as we had suggested, and in revised figure 3b, the authors now demonstrate that the mutant does indeed inhibit AcsA activity, albeit much more poorly than the AcuA-dC construct, which neither binds or acetylates AcsA. This data seems to argue that acetylation of AcsA is much more significant than the acetyl-CoA independent mode of inhibition. In light of this finding, the relevant experiment is to repeat the experiment shown in Figure 3a but with the AcuA_E102Q catalytic mutant. I suspect that the authors may have to go to higher ratios of AcuA to AcsA to see an effect on V_{max} or K_m . Having to go to an unusually high ratio may argue against the biological relevance of the acetyl-CoA independent mode of inhibition.

Reviewer #2

(Remarks to the Author)

Reviewer #3

(Remarks to the Author)

The revised manuscript, previously titled “Dual Inhibition through an Intertwined Acetyl-CoA Synthetase/Acetyltransferase Complex” and now titled “Regulation of Acetyl-CoA Biosynthesis via an Intertwined Acetyl-CoA Synthetase/Acetyltransferase Complex,” thoroughly addresses all concerns raised in the previous review. The authors have provided clear and comprehensive responses, supported by additional experiments and data, to resolve the issues highlighted earlier. Based on these comprehensive revisions, the manuscript has been significantly improved and is recommended for publication.

Concern raised in the previous review: Lack of quantitative evidence for the importance of acetyl-CoA-independent inhibition.

Response in the revised manuscript: The inhibition shown by the E102Q mutant in Fig. 3D provides evidence supporting the acetyl-CoA-independent inhibition mechanism.

Concern raised in the previous review: Lack of biological replicates and contradiction between HDX and Cryo-EM data.

Response in the revised manuscript: The authors repeated the HDX experiments with biological replicates and used the same protein construct for all tests. The new data align with the Cryo-EM results and resolve the earlier contradictions.

Concern raised in the previous review: Lack of binding affinity information for AcsA2:AcuA complex.

Response in the revised manuscript: The authors tried to measure K_d values by ITC experiments but couldn't get reliable results.

In the rebuttal letter, the authors explained that in the HDX-MS experiments they used AcuA/AcsA samples that were stable complexes, obtained through size-exclusion chromatography (SEC), not simple mixtures of AcuA and AcsA. While this supports the assumption of a single population, the low final concentration (5 μM after dilution in deuterated buffer) during the HDX reaction could cause the complex to dissociate during HDX experiments, potentially leading to multiple populations.

However, the authors noted that no bimodal distributions were observed in the mass envelopes of any HDX samples or data points, indicating that the HDX conditions likely reflect a single population.

Concern raised in the previous review: AcuA-acetyl-CoA complexation studies.

Response in the revised manuscript: The authors provided binding affinity data for the AcuA-acetyl-CoA system. The results confirm that HDX experimental conditions represent a predominantly AcuA-acetyl-CoA complexed population, ensuring the reliability of data.

Concern raised in the previous review: No HDX changes observed in key interaction regions.

Response in the revised manuscript: The new HDX experiments using a common protein construct and biological replicates, now show protection in key interaction regions, addressing this concern.

Reviewer #4

(Remarks to the Author)

Reviewer #5

(Remarks to the Author)

The authors responded to my suggestions (Reviewer #5), and have made appropriate changes to the manuscript, with one exception. I think they should better explain in the figure legend, the meaning of the y-axis is label, "Molar ratio", as they did in response to my comment.

Version 2:

Reviewer comments:

Reviewer #1

(Remarks to the Author)

The authors did a very nice job addressing our concerns and we now feel that the manuscript is suitable for publication.

Reviewer #2

(Remarks to the Author)

Response letter

We thank all reviewers for their constructive comments, which we have mostly addressed. This significantly improved our paper

Reviewer #1 (Remarks to the Author):

*In this study, Zheng, Bange and coworkers elucidate a novel inhibition mode of acetyl-CoA synthetase in *B. subtilis* bacteria. This inhibition occurs through the binding of an acetyl-transferase enzyme (AcuA), encoded in the same operon as AcsA, to AcsA. The authors investigate the molecular interaction between AcsA and AcuA using different techniques, including single particle Cryo-EM, Mass Photometry, Hydrogen-Deuterium Exchange Mass Spectrometry (HDXMS), site-directed mutagenesis and in vitro pulldown assays. They determine the structures of two complexes, the Acs dimer (Acs2) and an Acs dimer bound to one AcuA (Acs2-AcuA1). The cryo-EM dataset yielded 4 maps including the Acs2 dimer, the Acs2-AcuA1 trimer and 2 local refinement maps from the Acs2-AcuA1 complex of the CTD of Acs and the other of the AcuA complex. The C-terminus of AcuA binds near the Acs2 acetyl-CoA binding site, while AcsA K549 (the acetylation of which also inhibits AcsA activity) is found bound to AcuA near its active site. The HDX-MS data corroborates the binding interface found in the cryo-EM structure of Acs2-AcuA1 while the mass photometry data and activity assays show an acetyl-Coa dependent change in the oligomeric state of this complex.*

*Together, this study reveals a new acetyl-CoA independent mode of AcsA inhibition as an alternative to the acetyl-CoA dependent mode of AcuA-mediated acetylation of lysine 549 on AcsA to mediate AcsA inhibition. The studies are rigorous, well described and illustrated and nicely complements the recently published studies of Qin et al. (ref 23) that comes to similar conclusions using biochemical studies and Alphafold models. The studies have important implication for understanding the molecular mechanisms of the regulation of acetyl-CoA metabolism in *B. subtilis* with possible implications in other species.*

The authors should address the following points prior to publication:

Major points

1. Can the authors say anything about the significance of the structure of the AcsA dimer. Is there something interesting about the interface that provides functional relevance for why the protein forms a dimer?

We appreciate this question. The AcsA dimer is stabilized by hydrophobic and polar interactions, which may support the enzyme's structural integrity. The dimer interface also positions the active sites to potentially aid substrate binding and product release, though further studies are needed to confirm this.

2. Page 4 – lines 29-30, the authors refer to “the other mutations” that have no effect on the AcsA-AcuA interaction. Can the authors say something about (and illustrate) what these residues are doing.

The other mutations are located close to of AcsA_K549 and include negatively charged residues such as E85, E97, Y136, and T134 in AcuA, as well as positively charged residues R546 and S547 in AcsA. Following reviewer suggestion, we added a new supplementary figure (Supplementary Fig. 8) to illustrate the locations and potential roles of these residues within the AcsA-AcuA interface.

3. Figure 3a; The mixed type of inhibition (steric vs lysine 549 acetylation) that is observed in this figure would be easier to deconvolute if the authors titrated in a catalytic mutant of AcuA instead of the wildtype protein.

We appreciate this comment and constructed an AcuA_E102Q mutant recently shown to be catalytically inactive (Qin et al., 2024, Nature Communications). We could show that AcuA_E102Q still interacts with AcsA (Fig. 3b). Furthermore, our experiments demonstrated that AcuA_E102Q significantly inhibits AcsA activity, although its inhibitory effect was lower than that of the original protein used as control (Fig. 3c). These findings further support our model that a mixed type of inhibition is relevant for tight AcsA regulation.

4. Figure 3b; while I assume that that the AcuA_deltaC construct is unable to form a complex with AcsA it would be reassuring to confirm this using MP.

We avoided using GST-tagged proteins for MP analysis due to the presence of multiple species, which made them unsuitable. Instead, we designed a new construct, 6xHis-AcuA_ΔC, specifically for MP analysis. However, this protein showed a strong propensity to aggregate (Figure R1), rendering it unsuitable for the intended analysis.

Figure R1. Purification of N-His-AcuA_ΔC. (for reviewer only)

5. Figure 4; the authors should show the effect of Ac-CoA addition on AcsA dimer as a control.

We have included this control – Ac-CoA does not show an effect on the AcsA-dimer (Supplementary Fig. 11).

6. The physiological connection to acetate concentration described in the discussion is difficult to follow. The authors state “High levels of acetate are required to break up this complex demonstrating that it significantly adds to the burden of Acs activation.” But high levels of acetate would both activate Acs to produce more acetyl-CoA (and disrupt the AcuA complex) and also inhibit it by producing more acetyl-CoA product to acetylate Acs at K549 (via AcuA).

We agree with the reviewer that high levels of acetate increase acetyl-CoA production, leading to the disruption of the AcsA-AcuA complex. We have revised the text accordingly.

7. Were the authors able to generate a 3D class of the particles with the AcsA2-AcuA2 tetramer? It might be worthwhile to share an image of the other classes from the 3D classification step in

Supplementary Fig. 2. The authors could try re-picking particles based on an AlphaFold model of the tetramer complex to see if AcsA2-AcuA2 inhibits AcsA in the same way. Alternatively, the authors could use the Mass Photometry data to optimize the correct substrate and product concentrations to enrich the population of particles pertaining to specific oligomeric states.

Unfortunately, we were not able to identify AcsA2-AcuA2 in the dataset. Initial particle picking gives approximately 3 million particles. Extensive 2D and 3D classification showed high resolutions particles are either AcsA2 or AcsA2-AcuA. The image of 3D classification on 1,194,314 particles with T=4 is now added.

Figure R2. 2D and 3D classification of AcsA-AcuA complex

Minor points

8. Pg 3, lines 22-24; Use of T values in the results section is too technical for the non-specialist. If the authors want to use this in the results section, they need to define the significant and difference between T2 and T4.

We agree with the reviewer and moved the details to the method section.

10. Supplemental Fig. 4; the color coding is confusing. To be consistent with the text, it would make the most sense to just color-code the N- and C- term domains.

We have made the changes as suggested by the reviewer.

11. Throughout the text the authors use the term “properly resolved” but it is unclear to me why the word “properly” is used. It is resolved or not resolved.

12. Page 3, line 27: “homdimer” should be ‘homodimer’

13. Page 3, line 30: “PD-ID” should be “PDB ID”

14. Page 3 line 43; change “unknown conformation” to “unreported conformation”

15. Page 3, line 45: “AscA” should be “AcsA”

16. Page 4, line 8-9: “AcuA_F210” should be “AcuA_Y210”

We made these changes as requested.

17. Page 6, line 15-16; The sentence that reads “Acetate activated by Ac-CoA synthetases is probably the poorest source for Ac-CoA and thus this pathway needs to be efficiently paused if not absolutely required.” Is confusing. Can the authors reword or elaborate.

We rephrased the sentence accordingly.

Reviewer #2 (Remarks to the Author):

Reviewer #3 (Remarks to the Author):

Overview: The manuscript titled “Dual Inhibition through an Intertwined Acetyl-CoA Synthetase/Acetyltransferase Complex” explores the acetyl-CoA-independent inhibition of acetyl-CoA synthetase (AcsA) by acetyl-CoA acetyltransferase (AcuA), in addition to the known acetyl-CoA-dependent inhibition mechanism. This study integrates biochemical kinetics assays, Cryo-EM structural determination, and HDX-MS analysis to identify the binding interface and elucidate the complex regulatory interactions between AcsA and AcuA as basis for acetyl-CoA-independent inhibition. Much of the data are convincing and the discussion of possible regulatory mechanism is very interesting. However, a number of issues need to be addressed.

I- Lack of quantitative evidence for the importance of acetyl-CoA-independent inhibition: The manuscript could benefit from providing clearer evidence for acetyl-CoA-independent inhibition of acetyl-CoA synthetase. Based on the cryo-EM modeled complex of AcsA2, the paper shows an interaction between the C-terminal region of AcuA and the active site at the N-terminus of AcsA. Deletion of the C-terminal region (AcuA- Δ C) weakens the interaction between AcuA and AcsA but does not completely abolish complex formation, as seen in the GST-pulldown assay (Fig. 2h).

AcuA exhibits mixed inhibition of AcsA kinetics in the presence of acetyl-CoA-dependent inhibition (Fig. 3A). To establish the role of above interaction in the acetyl-CoA-independent inhibition of AcsA by AcuA, the paper demonstrates that deletion of the Δ C region changes the rate of product formation (Fig 3C). With this information, it was concluded that deletion of the C-terminus of AcuA weakens complex formation and hence may require higher amounts of protein to achieve the same level of inhibition.

Critique: In this context, it would have been very helpful to see quantitative inhibition kinetics data (impact on k_{cat} and K_m) conducted with varying levels of AcuA- Δ C, for comparison to the AcsA + AcuA conditions.

We followed the reviewers' suggestions and generated a new construct, 6xHis-AcuA_ΔC, to make kinetic assays for comparison with 6xHis-AcuA-WT, however, the protein aggregated – compare to our response to Rev 1. Therefore, we did not make a kinetic analysis. Instead, we tested the activity of AcsA in the presence of different concentrations of GST-AcuC-ΔC. No inhibition was observed (**Figure R3**).

Furthermore, to quantify acetyl-CoA-independent inhibition, we constructed GST_AcuA_E102Q, which is unable to catalyze the acetylation on lysine 549 anymore (Qin et al., 2024, Nature Communications), but can still form a complex with AcsA (**Fig. 3b**). Furthermore, HPLC analysis revealed that AcuA_E102Q significantly inhibits AcsA activity, although its inhibitory effect is not as strong as that of AcuA_WT (**Fig. 3c**).

Figure R3. Activity assays of AcsA in the presence of different amount of GST-AcuA-ΔC.

DeltaC3 means three times higher amount of DeltaC1

II-HDX-MS Data in the Manuscript: The HDX work presented in this manuscript demonstrates high sequence coverage and good repeatability in data collection. The authors utilized the same set of redundant peptides for both the apo-form and protein complexes, allowing for direct comparisons between different states. The peptide data from various samples are well-organized in structured tables, made available in PRIDE, which facilitates comparisons across experimental conditions. Additionally, the HDX protection analysis is tabulated and presented.

As per the suggestions below by this reviewer, we conducted further HDX-MS experiments and analysis, the results of which do replace all previous HDX-MS-related figures in the revised manuscript and supplementary dataset. These new data were uploaded at the PRIDE repository under the accession number PXD058390 and may be accessed via the following credentials:

Reviewer access details

Log in to the PRIDE website using the following details:

Project accession: PXD058390

Token: WZ1ZJZBepmEK

Alternatively, reviewer can access the dataset by logging in to the PRIDE website using the following account details:

Username: reviewer_pxd058390@ebi.ac.uk

Password: bgl33v328kKF

Furthermore, this reviewer pointed out an apparent bimodality in HDX under V. and VI., and more specifically differences in that bimodality between our AcuA/AcsA and AcuA ± Ac-CoA experiments with respect to AcuA. It turned out that the proteins employed before were AcuA-His6 and AcsA-His6 for the complex and His6-AcuA for the Ac-CoA binding experiment. Because of this usage of different AcuA constructs, the C-His6-tagged proteins appearing less stable in our hands and the C-His6-tag at AcuA potentially interfering with AcsA interaction, we modified our HDX-MS experiment of this revised version as follows:

- we performed all experiments in biological triplicates each consisting of technical triplicates
- included multiple charge states per peptide whenever possible
- we employed His6-AcuA and His6-AcsA in all of those experiments instead of the mixture of C-His6 and N-His6-tagged protein constructs

The consequences that these modifications result in are described below at the respective reviewer's points.

Critique: There are several points in the manuscript that require further clarification.
i) Lack of Binding Affinity Information (Kd): The HDX-MS experiments were conducted to identify the binding interface between homodimer of AcsA and AcuA (AcsA2:AcuA complex) that was identified in cryo-EM. The HDX-MS of AcsA:AcuA complex were conducted using equimolar concentrations (5 μM: 5 μM).

In the HDX-MS experiments elaborating on the AcuA/AcsA complex stock solutions of control samples contained 50 μM of the individual AcuA or AcsA proteins resulting in 5 μM final concentration during the HDX reaction (due to the 10-fold dilution in deuterated buffer to initiate HDX). However, the AcuA/AcsA complex sample was not a simple mixture of AcuA and AcsA but

instead a stable complex obtained after size-exclusion chromatography (see materials and methods section, compare to Fig. 1 for a representative SEC experiment).

The manuscript does not provide key binding affinity (K_d) values for the various interactions observed. As a result, without information on the binding strengths and relative populations of the different species (AcsA monomer, AcuA monomer, AcsA dimer, AcuA dimer, AcsA:AcuA AcsA2:AcuA and others), it is difficult to interpret the HDX traces with regard to the impact of the alternate protein.

We employed a SEC-purified AcuA/AcsA complex for our HDX-MS studies. This was established by mixing an excess of AcuA with AcsA (because the latter is the larger protein and thus the established complex easier to separate from non-complexed AcuA than AcsA) and collection of the resulting AcuA/AcsA-containing fractions shifted to higher molecular weight (see also Fig. 1 for a representative SEC experiment).

Furthermore, we performed the HDX-MS experiments as suggested by this reviewer in biological triplicates for further robustness of our assertions.

The fact that the protection offered by the binding partners disappears at the longer time points indicates significant amount of uncomplexed partners in most instances: protection can be detected at early times but disappears as the protein being assayed undergoes competing, unprotected exchange. The implication of relatively weak binding between AcsA and AcsU would be resolved with the availability of K_d values. Can the authors estimate K_d values from their kinetic assays (Figure 3)?

We tried to determine the binding affinity by ITC measurements. Unfortunately, our attempts remained not successful (see below).

Figure R4. The ITC measurement of AcuA vs AcsA. Ligands and proteins were prepared in buffer (20 mM HEPES, 20 mM MgCl₂, 20 mM KCl, 200 mM NaCl, pH 7.5). BsAcua (81–83 μM) and Acetyl-CoA (1 mM) or BsAcsA (32–52 μM) and BsAcua (475–596 μM) were analyzed using MicroCal PEAQ-ITC at 25 °C. Measurements involved 13 or 19 injections with 150 s spacing. Data were processed with PEAQ-ITC Analysis Software and plotted using GraphPad Prism.

Moreover, we are confident that our HDX data does not speak for a weak interaction.

Firstly, in all of our HDX-MS experiments (Fig. s6: AcsA in the AcuA/AcsA complex, Fig. s7: AcuA in the AcuA/AcsA complex, Fig. s11: acetyl-CoA binding to AcuA) the differences between the sample states shown in the panel C heat maps occur at different time-points of HDX and are not restricted to only to either the very first or later time-points.

Secondly, the sampled time-point of an HDX reaction at which a difference may occur between two sample states is not related to any fraction of uncomplexed sample, which would instead rather be apparent by bimodal behavior in HDX. Instead, the time-point of an HDX reaction at which a difference between two sample states may or may not occur is governed by the intrinsic HDX rate (or protection factor) of an amide proton. Amide protons in disordered areas exchange fast while those embedded in areas of higher-order structure exchange relatively slower. Hence, if a conformational change of the sample occurs in a disordered entity, we would expect to observe

differences in HDX between two samples in the earlier (e.g., 10 s) HDX time-points because upon longer deuteration any amide proton in that disordered area will exchange irrespective of whether it may be more protected upon binding of a ligand. Conversely, if an entity of the protein analyte containing slow exchanging amide protons undergoes a conformational change (if it is not the border case of complete unfolding of HOS) we would typically rather observe a difference in HDX at later time-points because only then an HDX at these amide protons will happen – sampled at earlier time-points these amide protons typically do not incorporate deuterium at all.

ii) AcuA-Acetyl-CoA Complexation Studies: It would also be helpful to include key binding affinity (K_d) values to better understand the relative populations of complexed and apo forms in the AcuA-Acetyl-CoA complexation studies.

We thank the reviewers' suggestion. We measure the binding affinity of AcuA to Ac-CoA by using ITC, the result showed that the K_d value is 9.47 ± 0.51 μM, see please (Supplementary Fig. 14)

This K_D also agrees with the range reported in literature, i.e., K_D of Ac-CoA binding to the *Pseudomonas aeruginosa* GNAT superfamily protein PA4794 determined by ITC with 3.8 ± 0.5 μM (PMID: 24003232) and a K_m of 22 ± 2 μM for Ac-CoA in enzymatic assays on *Bacillus subtilis* AcuA at saturating concentrations of an acceptor lysine-containing peptide (PMID: 18487328), which may serve as a good approximation given the experimental conditions employed in that manuscript.

In our HDX-MS experiments on Ac-CoA binding to *Bacillus subtilis* AcuA, we employed final concentrations of AcuA and Ac-CoA of 5 μM and 1 mM, respectively, during the HDX reaction. Based on our and literature K_d estimates, this should result in almost full saturation of AcuA with Ac-CoA during HDX-MS.

We added this information in the text as following sentence: “Based on our ITC measurements, acetyl-CoA exhibits a binding affinity to AcuA with a K_D of 9.47 ± 0.51 μM.”

iii) Lack of Biological Replicates: It appears that the HDX-MS data were obtained using technical triplicates, but no biological replicates are included. Incorporating biological replicates would strengthen the reliability and robustness of the conclusions, as biological variability, which can influence protein behaviour under different conditions, has not been accounted for.

We repeated all HDX-MS experiments in the manuscript in biological triplicates each consisting of technical triplicates (see Supplementary Dataset and dataset uploaded at the PRIDE repository)

and furthermore included multiple charge states per peptide whenever possible, both of which increases the robustness of the HDX-MS data presented.

The results of these revised experiments are consistent with the experimental data presented in the initial version of the manuscript except for two implications:

- We do not observe bimodality and differences thereof anymore for the two AcuA datasets, which was probably because of the usage of different protein constructs (see initial comment to this reviewer).
- Switching the His6-tags from C to N-termini, also because the AcuA-His6 version was probably interfering at least partially with the AcuA/AcsA complex, now reveals further areas of HDX changes, all of which are in good agreement with the cryo-EM structure.

iv) No Peptide-Specific Back-Exchange Correction: Performing peptide-specific back-exchange correction would help ensure the strength of the exchange measurements reported, especially when varying numbers of redundant peptides are used, as is the case in this work.

The HDX-MS experiments presented in this study contain side-by-side comparisons of different proteins and their complexes (experiment 1: AcuA, AcsA, AcuA/AcsA complex) and a ligand binding experiment (experiment 2: AcuA ± acetyl-CoA). For such comparative HDX-MS analyses the determination of peptide level back-exchange is not strictly necessary as the general back-exchange of the methodology (our instrument + reagent compositions) can be assumed to be comparable (see also Masson GR, 2019, Nat Methods)¹.

v) Contradiction Between HDX and Cryo-EM Data: The HDX analysis (summarized in Fig. 2E) shows that the C-terminal domain (CTD) of AcsA is less protected in the complex, implying that the CTD becomes unstructured upon complex formation. However, This observation appears to differ from the cryo-EM data, which shows that the CTD of AcsA becomes resolvable in the AcuA-bound complex, whereas it remains unresolved in the AcsA homodimer (page 3-line 39; Fig 2B, and 2C). More discussion is needed to reconcile this discrepancy.

In our initial data there were areas within the AcsA_CTD that, over the time course of HDX, underwent different exchange patterns and differences therein, i.e., an increased HDX in the complex at earlier time-points and a decreased HDX at later time-points. We do not agree that this would be entirely contradictory with the structure. The mixture of increased and decreased HDX within the very same peptide/area (in case of the AcsA_CTD primarily α -strands) could instead indicate subtle conformational changes within the housing α -strand central to the AcsA_CTD.

With the repetition of our experiments though, this behaviour was much less pronounced to a degree that it does not appear to be significant anymore for most areas it was observed before with the exception of AcsA residues 530-539 located in the AcsA_CTD still exhibiting higher HDX in context of the AcuA/AcsA complex.

vi) No HDX Changes Observed in Key Interaction Regions: The acetyl-CoA-independent inhibition proposed in this work is based on the interaction between the C-terminus of AcuA and the active site residues of the N-terminal domain (NTD) of AcsA. However, the HDX data for AcuA (Fig. S7C, peptide 203-210) show no significant protection factor changes upon complex formation. Similarly, the residues in the very end of the NTD of AcsA that interact with AcuA do not exhibit protection changes in the HDX data (Fig. S6C). These findings warrant further clarification to strengthen the proposed mechanism and its role in acetyl-CoA-independent inhibition.

This was not satisfactory to us too. We believe that the C-His6-tag at AcuA employed in our initial experiments impeded AcuA/AcsA complex formation to some extent; as evident from our cryo-EM structure the AcuA C-terminal tail extends into the AcsA active site. Hence, during revision of all HDX-MS experiments we employed N-His6-tagged AcuA and AcsA protein constructs instead of the C-His6-tagged versions used before.

As now apparent from the revised figures of this HDX-MS experiment (Supplementary Fig. 6: AcsA in the AcuA/AcsA complex, Supplementary Fig. 7: AcuA in the AcuA/AcsA complex), residues 301-317 of AcsA lining its catalytic center and the C-terminal AcuA residues 205-210 now exhibit HDX reduction in context of the complex (Note that both of these areas are covered by multiple peptides all showing the same behavior).

Reviewer #4 (Remarks to the Author):

Reviewer #5 (Remarks to the Author):

*The manuscript by Zheng et al., reports the characterization of the structure of a complex between acetyl-CoA synthetase (Acs) and a GCN5-related N-acetyltransferase (AcuA), and the effect of complex-formation on Acs catalytic activity. Acs catalyzes the formation of acetyl-CoA from acetate and CoA, a reaction that requires the hydrolysis of ATP to AMP plus PPi. The Acs-catalyzed reaction involves a catalytic Lysine residue, and in the case of the *Bacillus subtilis* Acs this is lysine residue-549. Acs enzymes from almost all biological sources can be inactivated by the acetylation of this catalytic lysine residue, and in *B. subtilis* this acetylation reaction is catalyzed by AcuA. The authors find that the formation of a Acs-AcuA complex is inhibitory to Acs catalysis, and this complex occurs in the absence of acetyl-CoA - thus they propose that this is an acetyl-CoA-independent mechanism for regulating Acs activity. Thus, two mechanism for regulating acetyl-CoA generation from acetate, one that is dependent on acetyl-CoA and another that is acetyl-CoA independent provides more stringent control.*

We sincerely appreciate your overall positive comments.

Suggests:

1) Strictly speaking the acetylation of the active site Lys residue is a “post-translational inactivation” of Acs, and I suggest that the authors use this phraseology rather than “inhibition of Acs”. One can thereby better distinguish the Acs-AcuA complex formation as an inhibitory mechanism from the “post-translational inactivation” by acetylation.

We agree the review for phrases. We have made the change accordingly.

2) However, enzymologically could not the formation of the AcuA-Acs complex be considered as the enzyme-substrate intermediate complex in the acetylation of Acs by AcuA reaction? With Acs being one of the substrates and AcuA being the enzyme. Subsequently the AcuA-Acs intermediate requires the second substrate of the acetylation reaction, namely acetyl-CoA. I suppose one could argue that the novelty is the stable formation of the enzyme-substrate complex formation (AcuA-Acs) in the absence of the second substrate. Then, the question becomes, is this physiologically

relevant in vivo and/or is this unique to the acetylation of Acs, or does it occur commonly in other protein acetylation reactions – for example, histone acetylation.

Thank you for the insightful comment. We agree that the AcuA-Acs complex could be viewed as an enzyme-substrate intermediate, with AcuA acting as the enzyme and Acs as the substrate. We consider the formation of this stable complex in the absence of acetyl-CoA as an intriguing finding. In the revised paper we introduce the AcuA_E102Q mutant, which cannot catalyze the acetylation of AcsA_K549, but still forms a complex with AcsA. It also inhibits AcsA (**Figs. 3b and 3c**). We think that it is an interesting phenomenon that AcsA has its deactivator always close, especially because the proteins are encoded in an operon and occur in similar amounts.

The reviewer's question about the physiological relevance of this mechanism and its potential occurrence in other acetylation systems, such as histone acetylation, merits further investigation but appears outside the scope of the current manuscript.

3) Somewhat related to the above, on p5, lines 8-18, the authors find that the C-terminal 7-residues of AcuA mediates the inhibition of Acs by binding to the CoA-binding site of Acs. But I notice that in the experiments shown in Figure 3c, the synthetic C-terminal 7-residue peptide of AcuA that causes 90% inhibition of Acs activity is used at a 1mM concentration. This seems rather high concentration. Is this a physiological concentration? What is the concentration of Acs in these experiments?

We agree with the reviewer that 1 mM is rather high concentration. However, in aggregate with our data on AcuA_ΔC and AcuA_E102Q we are confident that non-enzymatic inhibition is important and that the AcuA C-terminus inserting into the substrate binding pocket of AcsA is part of this regulatory scheme.

Minor points:

a) Figure 4 - The y-axis is labeled as "Molar ratio", and this number is presented as a percentage. But it's unclear what ratio was measured? Acs:AcuA? What is 100%? Please better define?

We thank the reviewer for highlighting this point. The 100% refers to the total counts recorded by MP within one minute. The ratio represents the N counts (peak of interest) / N counts (total).

b) The first sentence of the Discussion, the authors state, "Acetate activated by Ac-CoA synthetases is probably the poorest source for Ac-CoA and thus this pathway needs to be efficiently paused if not absolutely required." It's not clear what "poorest" means. Could they better define - maybe they are referring to the most "energy expensive" reaction, as the Acs reaction

costs 2 molecules of ATP, as ATP is hydrolyzed to AMP, and the reaction needs to be pulled to the acetyl-CoA product side, by the hydrolysis of the other product, pyrophosphate to phosphate.

We agree with the reviewer and rephrase accordingly: "Acetate, when activated by Ac-CoA synthetases, is an energy expensive source of Ac-CoA. Therefore, this pathway should be effectively paused unless it is absolutely necessary."

REVIEWER COMMENTS

Reviewer #1 (Remarks to the Author):

The authors did a nice job addressing our concerns, but one important point still remains about the significance of the acetyl-CoA independent inhibition of AcsA by AcuA. This is relevant to the following comments that we made below:

Figure 3a; The mixed type of inhibition (steric vs lysine 549 acetylation) that is observed in this figure would be easier to deconvolute if the authors titrated in a catalytic mutant of AcuA instead of the wildtype protein.

Related to our comment, Reviewer 5 (and also a related point of review 3) bring up a similar concern, noting the possibility that the acetyl-CoA independent inhibition might be an enzyme intermediate as opposed to an independent mode of inhibition.

To address this concern, the authors prepared a AcuA_E102Q catalytic mutant, as we had suggested, and in revised figure 3b, the authors now demonstrate that the mutant does indeed inhibit AcsA activity, albeit much more poorly than the AcuA-dC construct, which neither binds or acetylates AcsA. This data seems to argue that acetylation of AcsA is much more significant than the acetyl-CoA independent mode of inhibition. In light of this finding, the relevant experiment is to repeat the experiment shown in Figure 3a but with the AcuA_E102Q catalytic mutant. I suspect that the authors may have to go to higher ratios of AcuA to AcsA to see an effect on Vmax or Km. Having to go to an unusually high ratio may argue against the biological relevance of the acetyl-CoA independent mode of inhibition.

We thank the reviewer for their overall positive comments and insightful suggestions.

Following the reviewer's smart recommendation, we performed kinetic assays corroborating our findings that the E102Q variant of AcuA is a steric inhibitor of AcsA (at similar concentrations as the wildtype protein). We have updated the manuscript accordingly. The data are shown in Supplementary Fig S9, and commented in the updated manuscript.

Reviewer #2 (Remarks to the Author):

We thank the reviewer for their time and effort in reviewing our work.

Reviewer #3 (Remarks to the Author):

The revised manuscript, previously titled "Dual Inhibition through an Intertwined Acetyl-CoA Synthetase/Acetyltransferase Complex" and now titled "Regulation of Acetyl-CoA Biosynthesis via an Intertwined Acetyl-CoA Synthetase/Acetyltransferase Complex," thoroughly addresses all concerns raised in the previous review. The authors have provided clear and comprehensive responses, supported by additional experiments and data, to resolve the issues highlighted earlier. Based on these comprehensive revisions, the manuscript has been significantly improved and is recommended for publication.

Concern raised in the previous review: Lack of quantitative evidence for the importance of acetyl-CoA-independent inhibition.

Response in the revised manuscript: The inhibition shown by the E102Q mutant in Fig. 3D provides evidence supporting the acetyl-CoA-independent inhibition mechanism.

Concern raised in the previous review: Lack of biological replicates and contradiction between HDX and Cryo-EM data.

Response in the revised manuscript: The authors repeated the HDX experiments with biological replicates and used the same protein construct for all tests. The new data align with the Cryo-EM results and resolve the earlier contradictions.

Concern raised in the previous review: Lack of binding affinity information for AcsA2:AcuA complex. Response in the revised manuscript: The authors tried to measure K_d values by ITC experiments but couldn't get reliable results.

In the rebuttal letter, the authors explained that in the HDX-MS experiments they used AcuA/AcsA samples that were stable complexes, obtained through size-exclusion chromatography (SEC), not simple mixtures of AcuA and AcsA. While this supports the assumption of a single population, the low final concentration (5 μM after dilution in deuterated buffer) during the HDX reaction could cause the complex to dissociate during HDX experiments, potentially leading to multiple populations.

However, the authors noted that no bimodal distributions were observed in the mass envelopes of any HDX samples or data points, indicating that the HDX conditions likely reflect a single population.

Concern raised in the previous review: AcuA-acetyl-CoA complexation studies. Response in the revised manuscript: The authors provided binding affinity data for the AcuA-acetyl-CoA system. The results confirm that HDX experimental conditions represent a predominantly AcuA-acetyl-CoA complexed population, ensuring the reliability of data.

Concern raised in the previous review: No HDX changes observed in key interaction regions. Response in the revised manuscript: The new HDX experiments using a common protein construct and biological replicates, now show protection in key interaction regions, addressing this concern.

We sincerely appreciate the reviewer's positive comments and insightful suggestions.

Reviewer #4 (Remarks to the Author):

We sincerely thank the reviewer for their time and effort in reviewing our work.

Reviewer #5 (Remarks to the Author):

The authors responded to my suggestions (Reviewer #5), and have made appropriate changes to the manuscript, with one exception. I think they should better explain in the figure legend, the meaning of the y-axis is label, "Molar ratio", as they did in response to my comment.

We thank the reviewer for their overall positive comments on the revised manuscript. We have updated the figure legend accordingly.